# M-L2O: Towards Generalizable Learning-to-Optimize by Test-Time Fast Self-Adaptation

**Junjie Yang**[1], **Xuxi Chen**[2], **Tianlong Chen**[2], **Zhangyang Wang**[2], **Yingbin Liang**[1]

[1]The Ohio State University, [2]University of Texas at Austin

`{yang.4972,liang.889}@osu.edu`
`{xxchen,tianlong.chen,atlaswang}@utexas.edu`

## Abstract

Learning to Optimize (L2O) has drawn increasing attention as it often remarkably accelerates the optimization procedure of complex tasks by "overfitting" specific task types, leading to enhanced performance compared to analytical optimizers. Generally, L2O develops a parameterized optimization method (*i.e.*, "optimizer") by learning from solving sample problems. This data-driven procedure yields L2O that can efficiently solve problems similar to those seen in training, that is, drawn from the same "task distribution". However, such learned optimizers often struggle when new test problems come with a substantial deviation from the training task distribution. This paper investigates a potential solution to this open challenge, by meta-training an L2O optimizer that can perform fast *test-time self-adaptation* to an out-of-distribution task, in only a few steps. We theoretically characterize the generalization of L2O, and further show that our proposed framework (termed as **M-L2O**) provably facilitates rapid task adaptation by locating well-adapted initial points for the optimizer weight. Empirical observations on several classic tasks like LASSO, Quadratic and Rosenbrock demonstrate that M-L2O converges significantly faster than vanilla L2O with only 5 steps of adaptation, echoing our theoretical results. Codes are available in https://github.com/VITA-Group/M-L2O.

## 1 Introduction

Deep neural networks are showing overwhelming performance on various tasks, and their tremendous success partly lies in the development of analytical gradient-based optimizers. Such optimizers achieve satisfactory convergence on general tasks, with manually-crafted rules. For example, SGD (Ruder, 2016) keeps updating towards the direction of gradients and Momentum (Qian, 1999) follows the smoothed gradient directions. However, the reliance on such fixed rules can limit the ability of analytical optimizers to leverage task-specific information and hinder their effectiveness.

Learning to Optimize (L2O), an alternative paradigm emerges recently, aims at *learning* optimization algorithms (usually parameterized by deep neural networks) in a data-driven way, to achieve faster convergence on specific *optimization task* or **optimizee**. Various fields have witnessed the superior performance of these learned **optimizers** over analytical optimizers (Cao et al., 2019; Lv et al., 2017; Wichrowska et al., 2017; Chen et al., 2021a; Zheng et al., 2022). Classic L2Os follow a two-stage pipeline: at the *meta-training* stage, an L2O optimizer is trained to predict updates for the parameters of optimizees, by learning from their performance on sample tasks; and at the *meta-testing* stage, the L2O optimizer freezes its parameters and is used to solve new optimizees. In general, L2O optimizers can efficiently solve optimizees that are similar to those seen during the meta-training stage, or are drawn from the same "task distribution".

However, new unseen optimizees may substantially deviate from the training task distribution. As L2O optimizers predict updates to variables based on the dynamics of the optimization tasks, such as gradients, different task distributions can lead to significant dissimilarity in task dynamics. Therefore, L2O optimizers often incur inferior performance when faced with these distinct unseen optimizees.

Such challenges have been widely observed and studied in related fields. For example, in the domain of meta-learning (Finn et al., 2017; Nichol & Schulman, 2018), we aim to enable neural networks to

be fast adapted to new tasks with limited samples. Among these techniques, Model-Agnostic Meta Learning (MAML) (Finn et al., 2017) is one of the most widely-adopted algorithms. Specifically, in the meta-training stage, MAML makes inner updates for individual tasks and subsequently conducts back-propagation to aggregate the gradients of individual task gradients, which are used to update the meta parameters. This design enables the learned initialization (meta parameters) to be sensitive to each task, and well-adapted after few fine-tuning steps.

Motivated by this, we propose a novel algorithm, named M-L2O, that incorporates the meta-adaption design in the meta-training stage of L2O. In detail, rather than updating the L2O optimizer directly based on optimizee gradients, M-L2O introduces a nested structure to calculate optimizer updates by aggregating the gradients of meta-updated optimizees. By adopting such an approach, M-L2O is able to identify a well-adapted region, where only a few adaptation steps are sufficient for the optimizer to generalize well on unseen tasks. In summary, the contributions of this paper are outlined below:

- To address the unsatisfactory generalization of L2O on out-of-distribution tasks, we propose to incorporate a meta adaptation design into L2O training. It enables the learned optimizer to locate in well-adapted initial points, which can be fast adapted in only a few steps to new unseen optimizees.

- We theoretically demonstrate that our meta adaption design grants M-L2O optimizer faster adaption ability in out-of-distribution tasks, shown by better generalization errors. Our analysis further suggests that training-like adaptation tasks can yield better generalization performance, in contrast to the common practice of using testing-like tasks. Such theoretical findings are further substantiated by the experimental results.

- Extensive experiments consistently demonstrate that the proposed M-L2O outperforms various baselines, including vanilla L2O and transfer learning, in terms of the testing performance within a small number of steps, showing the ability of M-L2O to promptly adapt in practical applications.

## 2 RELATED WORKS

### 2.1 LEARNING TO OPTIMIZE

Learning to Optimize (L2O) captures optimization rules in a data-driven way, and the learned optimizers have demonstrated success on various tasks, including but not limited to black-box (Chen et al., 2017), Bayesian (Cao et al., 2019), minimax optimization problems (Shen et al., 2021), domain adaptation (Chen et al., 2020b; Li et al., 2020), and adversarial training (Jiang et al., 2018; Xiong & Hsieh, 2020). The success of L2O is based on the parameterized optimization rules, which are usually modeled through a long short-term memory network (Andrychowicz et al., 2016), and occasionally as multi-layer perceptrons (Vicol et al., 2021). Although the parameterization is practically successful, it comes with the "curse" of generalization issues. Researchers have established two major directions for improving L2O generalization ability: the first focuses on the generalization to similar optimization tasks but *longer* training iterations. For example, Chen et al. (2020a) customized training procedures with curriculum learning and imitation learning, and Lv et al. (2017); Li et al. (2020) designed rich input features for better generalization. Another direction focuses on the generalization to different optimization tasks: Chen et al. (2021b) studied the generalization for LISTA network on unseen problems, and Chen et al. (2020c) provided theoretical understandings to hybrid deep networks with learned reasoning layers. In comparison, our work theoretically studies general L2O and our proposals generalization performance under task distribution shifts.

### 2.2 FAST ADAPTATION

Fast adaptation is one of the major goals in the meta-learning area Finn et al. (2017); Nichol & Schulman (2018); Lee & Choi (2018) which often focuses on generalizing to new tasks with limited samples. MAML Finn et al. (2017), a famous and effective meta-learning algorithm, utilizes the nested loop for meta-adaption. Following this trend, numerous meta-learning algorithms chose to compute meta updates more efficiently. For example, FOMAML (Finn et al., 2017) only updated networks by first-order information; Reptile (Nichol & Schulman, 2018) introduced an extra intermediate variable to avoid Hessian computation; HF-MAML (Fallah et al., 2020) approximated the one-step meta update by Hessian-vector production; and Ji et al. (2022) adopted a multi-step approximation in updates. Meanwhile, many researchers designed algorithms to compute meta updates more wisely. For example, ANIL (Raghu et al., 2020) only updated the head of networks in the inner loop; HSML (Yao et al., 2019) tailored the transferable knowledge to different tasks; and MT-net (Lee & Choi, 2018) enabled meta-learner to learn on each layer's activation space. In terms of theories,

(Fallah et al., 2021) measured the generalization error of MAML; Fallah et al. (2020) captured the single inner step MAML convergence rate by Hessian vector approximation. Furthermore, Ji et al. (2022) characterized the multiple-step MAML convergence rate. Recently, LFT Zhao et al. (2022) combines the meta-learning design in Learning to Optimize and demonstrates its better performance for adversarial attack applications.

## 3 PROBLEM FORMULATION AND ALGORITHM

In this section, we firstly introduce the formulation of L2O, and subsequently propose M-L2O for generalizable self-adaptation.

### 3.1 L2O PROBLEM DEFINITION

Most machine learning algorithms adopt analytical optimizer, *e.g.* SGD, to compute parameter updates for general loss functions (we call it **optimizee** or **task**). Instead, L2O aims to estimate such updates by a model (usually a neural network), which we call **optimizer**. Specifically, the L2O optimizer takes the optimizee information (such as loss values and gradients) as input and generates updates to the optimizee. In this work, our objective is to learn the initialization of the L2O optimizer on training optimizees and subsequently finetune it on adaptation optimizees. Finally, we apply such an adapted

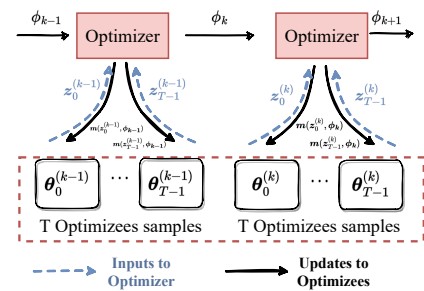

Figure 1: The pipeline of L2O problems.

optimizer to optimize the testing optimizees and evaluate their performance.

We define $l(\theta; \xi)$ as the loss function where $\theta$ is the **optimizee**'s parameter, $\xi = \{\xi_j (j = 1, 2, \ldots, N)\}$ denotes the data sample, then the **optimizee** empirical and population risks are defined as below:

$$\hat{l}(\theta) = \frac{1}{N} \sum_{j=1}^{N} l(\theta; \xi_j), \quad l(\theta) = \mathbb{E}_\xi l(\theta; \xi).$$

In L2O, the **optimizee**'s parameter $\theta$ is updated by the **optimizer**, an update rule parameterized by $\phi$ and we formulate it as $m(z_t(\theta_t; \zeta_t), \phi)$. Specifically, $z_t = z_t(\theta_t; \zeta_t)$ denotes the optimizer model input. It captures the $t$-th iteration's optimizee information and parameterized by $\theta_t$ with the data batch $\zeta_t$. Then, the update rule of $\theta$ in $t$-th iteration is shown as below:

$$\theta_{t+1}(\phi) = \theta_t(\phi) + m(z_t(\theta_t; \zeta_t), \phi) = \theta_t(\phi) + m(z_t, \phi). \tag{1}$$

The above pipeline is also summarized in Figure 1 where $k$ denotes the update epoch of the optimizer. Note that $\zeta_t$ refers to the data batch of size $N$ used at $t$-th iteration while $\xi_j$ refers to the $j$-th single sample in data batch $\xi$. For theoretical analysis, we only consider taking the optimizee's gradients as input to the optimizer for update, i.e., $z_t(\theta_t; \zeta_t) = \nabla_\theta \hat{l}(\theta_t(\phi))$. Therefore, the optimizee's gradient at $T$-th iteration over the optimizer $\nabla_\phi \theta_T(\phi)$ takes a form of

$$\nabla_\phi \theta_T(\phi) = \sum_{i=0}^{T-1} (\prod_{j=i+1}^{T-1} (I + \nabla_1 m(\nabla_\theta \hat{l}(\theta_{T+i-j}), \phi) \nabla_\theta^2 \hat{l}(\theta_{T+i-j})) \nabla_2 m(\nabla_\theta \hat{l}(\theta_i), \phi)), \tag{2}$$

where we assume that $\phi$ is independent from the optimizee's initial parameter $\theta_0$ and all samples are independent. The detailed derivation is shown in Lemma 1 in the Appendix.

Next, we consider a common initial parameter $\theta_0$ for all optimizees and define the **optimizer** empirical and population risks w.r.t. $\phi$ as below:

$$\hat{g}_t(\phi) = \hat{l}(\theta_t(\phi)), \quad g_t(\phi) = \mathbb{E}_\xi l(\theta_t(\phi); \xi) = l(\theta_t(\phi)), \tag{3}$$

where $\theta_t(\phi)$ updates in such a fashion: $\theta_{t+1}(\phi) = \theta_t(\phi) + m(z_t(\theta_t; \zeta_t), \phi)$. Typically, the L2O optimizer is evaluated and updated after updating the optimizees for $T$-th iterations. Therefore, the optimal points of the optimizer risks in Equation (3) are defined as below:

$$\widetilde{\phi}_* = \arg\min_\phi \hat{g}_T(\phi), \quad \phi_* = \arg\min_\phi g_T(\phi). \tag{4}$$

## 3.2 M-L2O ALGORITHM

To improve the generalization ability of L2O, we aim at learning a well-adapted optimizer initialization by introducing a nested meta-update architecture. Instead of directly updating $\phi$, we adapt it for one step (namely an inner update) and define a new empirical risk for the optimizer as follows:

$$\hat{G}_T(\phi) = \hat{g}_T(\phi - \alpha \nabla_\phi \hat{g}_T(\phi)), \tag{5}$$

where $\alpha$ is the step size for inner updates. Consequently, the optimal point of the corresponding updated optimizer is:

$$\widetilde{\phi}_{M*} = \arg\min_\phi \hat{G}_T(\phi). \tag{6}$$

Based on such an optimizer loss, we introduce M-L2O in Algorithm 1. Such a nested update design has been proved effective in the field of meta-learning, particularly MAML (Finn et al., 2017). Note that Algorithm 1 only returns the well-adapted optimizer initial point, which would require further adaptation in practice. We first denote the optimizees for training, adaptation, and testing by $g^1(\phi)$, $g^2(\phi)$, and $g^3(\phi)$, respectively, to distinguish tasks seen in different stages. Next, we obtain the results of meta training, denoted by $\widetilde{\phi}_{MK}^1$, via Algorithm 1, and we further adapt it based on $\hat{g}_T^2(\phi)$. The testing loss of M-L2O can be expressed as follows:

$$g_T^3(\widetilde{\phi}_{MK}^1 - \alpha \nabla_\phi \hat{g}_T^2(\widetilde{\phi}_{MK}^1)), \tag{7}$$

where $g_T^3(\phi)$ denotes the meta testing loss. Note that $\hat{g}$ refers to empirical risk and $g$ refers to population risk.

---

**Algorithm 1** Our Proposed M-L2O.

---

1: **Input:** Inner step size $\alpha$, Outer learning stepsize $\beta_k$, Total epochs $K$, Epoch number per task $S$, Optimizer initial point $\widetilde{\phi}_{M0}^1$, Training task $\hat{g}^1$, Adaptation task $\hat{g}^2$, Testing task $g^3$
2: **for** $k = 0, 1, \ldots, K-1$ **do**
3:     **if** $\mathrm{mod}(k, S) = 0$: $\theta_0(\widetilde{\phi}_{Mk}^1) = \theta_0$ (random initial) **else** $\theta_0(\widetilde{\phi}_{Mk}^1) = \theta_T(\widetilde{\phi}_{M(k-1)}^1)$
4:     **for** $t = 0, 1, \ldots, T-1$ **do**
5:         $\theta_{t+1}(\widetilde{\phi}_{Mk}^1) = \theta_t(\widetilde{\phi}_{Mk}^1) + m(\nabla_\theta \hat{l}_1(\theta_t(\widetilde{\phi}_{Mk}^1)))$      Note: $\hat{l}_1(\theta_t(\widetilde{\phi}_{Mk}^1)) = \hat{g}_t^1(\widetilde{\phi}_{Mk}^1)$
6:     **end for**
7:     Compute $\hat{G}_T^1(\widetilde{\phi}_{Mk}^1) = \hat{g}_T^1(\widetilde{\phi}_{Mk}^1 - \alpha \nabla_\phi \hat{g}_T^1(\widetilde{\phi}_{Mk}^1))$
8:     **update:** $\widetilde{\phi}_{M(k+1)}^1 = \widetilde{\phi}_{Mk}^1 - \beta_k \nabla_\phi \hat{G}_T^1(\widetilde{\phi}_{Mk}^1)$
9: **end for**
10: **Output:** $\widetilde{\phi}_{MK}^1$     **Testing Loss:** $g_T^3(\widetilde{\phi}_{MK}^1 - \alpha \nabla_\phi \hat{g}_T^2(\widetilde{\phi}_{MK}^1))$

---

# 4 GENERALIZATION THEOREM OF M-L2O

In this section, we introduce several assumptions and characterize M-L2O's generalization ability.

## 4.1 TECHNICAL ASSUMPTIONS AND DEFINITIONS

To characterize the generalization of meta adaptation, it is necessary to make strong convexity assumptions which have been widely observed (Li & Yuan, 2017; Du et al., 2019) under over-parameterization condition and adopted in the geometry of functions (Fallah et al., 2021; Finn et al., 2019; Ji et al., 2021).

**Assumption 1.** *We assume that the function $g_T^1(\phi)$ is $\mu-$strongly convex. This assumption also holds for stochastic $\hat{g}_T^1(\phi)$.*

To capture the relationship between the L2O function $\hat{g}_T(\phi)$ and the M-L2O function $\hat{G}_T(\phi)$, we make the following optimal point assumptions.

**Assumption 2.** *We assume there exists a non-trivial optimizer optimal point $\widetilde{\phi}_{M*}^1$ which is defined in Equation (6). Non-trivial means that $\nabla_\phi \hat{g}_T^1(\widetilde{\phi}_{M*}^1) \neq 0$ and $\widetilde{\phi}_{M*}^1 \neq \widetilde{\phi}_*^1$. Then, based on the definition of $\widetilde{\phi}_{M*}^1$ and the existence of trivial solutions, for any $\widetilde{\phi}_{M*}^1$, we have the following equation:*

$$\widetilde{\phi}_*^1 = \widetilde{\phi}_{M*}^1 - \alpha \nabla_\phi \hat{g}_T^1(\widetilde{\phi}_{M*}^1),$$

*where $\alpha$ is the step size for inner update of the optimizer. Note that we have defined the strongly-convex of landscape in Assumption 1 which validates the uniqueness of $\widetilde{\phi}^1_*$.*

The aforementioned assumption claims that there exist meta optimal points which are different from original task optimal points. Such an assumption is natural in experimental view. In MAML experiments where a single training task is considered, it is reasonable to expect that the solution would converge towards a well-adapted point instead of the task optimal point. Otherwise, MAML would be equivalent to simple transfer learning, where the learned point may not generalize well to new tasks.

**Assumption 3.** *We assume that $l_i(\theta)$ is $M$-Lipschitz, $\nabla l_i(\theta)$ is $L$-Lipschitz and $\nabla^2 l_i(\theta)$ is $\rho$-Lipschitz for each loss function $l_i(\theta)(i = 1, 2, 3)$. This assumption also holds for stochastic $\hat{l}_i(\theta)$, $\nabla\hat{l}_i(\theta)$ and $\nabla^2_\theta\hat{l}_i(\theta)(i = 1, 2, 3)$. We further assume that $m(z, \phi)$ is $M_{m1}$-Lipschitz w.r.t. $z$, $M_{m2}$-Lipschitz w.r.t. $\phi$ and $\nabla^2_\phi\theta_T(\phi)$ is $\rho_\theta$-Lipschitz.*

The above Lipschitz assumptions are widely adopted in previous optimization works (Fallah et al., 2021; 2020; Ji et al., 2022; Zhao et al., 2022). To characterize the difference between tasks for meta training and meta adaptation, we define $\Delta_{12}$ and $\widetilde{\Delta}_{12}$ as follows:

**Assumption 4.** *We assume there exist union bounds $\Delta_{12}$ and $\widetilde{\Delta}_{12}$ to capture the gradient and Hessian differences respectively between meta training task and adaptation task:*

$$\Delta_{12} = \max_\theta \|\nabla_\theta\hat{l}_1(\theta) - \nabla_\theta\hat{l}_2(\theta)\|, \qquad \widetilde{\Delta}_{12} = \max_\theta \|\nabla^2_\theta\hat{l}_1(\theta) - \nabla^2_\theta\hat{l}_2(\theta)\|.$$

Such an assumption has been made similarly in MAML generalization works (Fallah et al., 2021).

## 4.2 MAIN THEOREM

In this section, we theoretically analyze the generalization error of M-L2O and compare it with the vanilla L2O approach (Chen et al., 2017). Firstly, we characterize the difference of optimizee gradients between any two tasks ($\nabla_\phi\theta^1_T(\phi)$ and $\nabla_\phi\theta^2_T(\phi)$) in the following form:

**Proposition 1.** *Based on Equation (2), Assumptions 3 and 4, we obtain*

$$\|\nabla_\phi\theta^1_T(\phi) - \nabla_\phi\theta^2_T(\phi)\| \leq \sum_{i=0}^{T-1}(Q^{T-i-1}\Delta_{Ci} + M_{m2}Q^{T-i-2}\sum_{j=i+1}^{T-1}\Delta_{Dj}),$$

*where $\Delta_{Ci} = \mathcal{O}(Q^i\Delta_{12})$, $\Delta_{Dj} = \mathcal{O}(Q^j\Delta_{12} + \widetilde{\Delta}_{12})$ and $Q^i = (1 + M_{m1}L)^i$. Furthermore, we characterize the task difference of optimizer gradient $\nabla_\phi\hat{g}_T(\phi)$ as follows:*

$$\|\nabla_\phi\hat{g}^1_T(\phi) - \nabla_\phi\hat{g}^2_T(\phi)\| = \mathcal{O}(TQ^{T-1}\widetilde{\Delta}_{12} + Q^{2T-1}\Delta_{12}),$$

*where $Q = 1 + M_{m1}L$, $\Delta_{12}$ and $\widetilde{\Delta}_{12}$ are defined in Assumption 4.*

Proposition 1 shows that the difference in optimizer gradient landscape scales exponentially with the optimizee iteration number $T$. Specifically, it involves the $Q^{2T-1}$ term with gradient difference $\Delta_{12}$ and the $TQ^{T-1}$ term with Hessian difference $\widetilde{\Delta}_{12}$. Clearly, $Q^{2T-1}\Delta_{12}$ dominates when $T$ increases, which implies that the gradient gap between optimizees is the key component of the difference in optimizer gradient.

**Theorem 1.** *Suppose that Assumptions 1, 2, 3 and 4 hold. Considering Algorithm 1 and Equation (7), if we define $\delta_{13} = \|\phi^1_* - \phi^3_*\|$, set $\alpha \leq \min\{\frac{1}{2L}, \frac{\mu}{8\rho_{g_T}M_{g_T}}\}$, $\beta_k = \min(\beta, \frac{8}{\mu(k+1)})$ for $\beta \leq \frac{8}{\mu}$. Then, with a probability at least $1 - \delta$, we obtain*

$$\mathbb{E}[g^3_T(\widetilde{\phi}^1_{MK} - \alpha\nabla_\phi\hat{g}^2_T(\widetilde{\phi}^1_{MK})) - g^3_T(\phi^3_*)]$$

$$\leq (M_{g_T}(1 + L_{g_T}\alpha))\left\|\mathcal{O}(1)\frac{M_{g_T}}{\mu^2}\sqrt{\frac{L_{g_T} + \rho_{g_T}\alpha M_{g_T}}{\beta K} + \frac{M_{g_T}}{\beta\sqrt{K}}}\right\| + M_{g_T}\left\|\frac{2\sqrt{2}M_{g_T}}{\mu\sqrt{\delta N}}\right\| + M_{g_T}\delta_{13}$$

$$+ M_{g_T}\alpha\mathcal{O}(TQ^{T-1}\widetilde{\Delta}_{12} + Q^{2T-1}\Delta_{12}),$$

*where $Q = 1 + M_{m1}L$, $M_{g_T} = \mathcal{O}(Q^{T-1})$, $L_{g_T} = \mathcal{O}(TQ^{T-2} + Q^{2T-2})$, $\rho_{g_T} = \mathcal{O}(TQ^{2T-3} + Q^{3T-3})$, $K$ is total epoch number for meta training, $N$ is the batch size for optimizer training.*

To provide further understanding of generalization errors, we first make the following remark:

**Remark 1** (The choice of $\alpha$). *In Theorem 1, we set $\alpha \leq \frac{\mu}{8\rho_{g_T} M_{g_T}} = \mathcal{O}(\frac{1}{TQ^{3T-4}+Q^{4T-4}})$, thus the error term $M_{g_T}\alpha\mathcal{O}(TQ^{T-1}\widetilde{\Delta}_{12} + Q^{2T-1}\Delta_{12})$ vanishes with larger $T$. If we fix the iteration number $T$, then such an error term is determined by the gradient bound $\Delta_{12}$ and the Hessian bound $\widetilde{\Delta}_{12}$.*

The key components that lead to $Q$ dependency are the Lipschitz properties to characterize the L2O loss landscape, *e.g.* the Lipschitz term $L_{gT} = \mathcal{O}(TQ^{T-2} + Q^{2T-2})$ defined for $\nabla_\phi \hat{g}_T(\phi)$. The reason is due to our nested update procedure $\theta_{t+1}(\phi) = \theta_t(\phi) + m(\nabla_\theta \hat{l}(\theta_t(\phi)), \phi)$. If we take the gradient of the last update term $\hat{g}_T(\phi) = \hat{l}(\theta_T(\phi))$ over $\phi$, then it requires us to compute gradients iteratively for all $t = 0, 1, \ldots, T-1$, which leads to the exponential term.

Consequently, it can be observed that the generalization error of M-L2O can be decomposed into three components: (i) The first term determined by $\sqrt{\frac{L_{gT}+\rho_{g_T}\alpha M_{g_T}}{\beta K}} + \frac{M_{g_T}}{\beta\sqrt{K}}$ is dominated by the training epoch $K$. Such an error term characterizes how meta training influences the generalization; (ii) The second term $\|\frac{2\sqrt{2}M_{g_T}}{\mu\sqrt{\delta N}}\|$ reflects the empirical error introduced by limited samples; hence it is controlled by the sample size $N$. (iii) The last two error terms capture task differences. Specifically, $\delta_{13}$ measures the gap between training and testing optimal points, while $\Delta_{12}$ and $\widetilde{\Delta}_{12}$, which dominate the last error term and represent the gradient and Hessian union bounds, respectively, reflect the geometry difference between training and adaptation tasks.

For better comparison with L2O, we make the following remark about generalization of M-L2O and Transfer Learninig.

**Remark 2** (Comparison with Transfer Learning). *We can rewrite the generalization error of M-L2O in Theorem 1 in the following form:*

$$g_T^3(\widetilde{\phi}_{MK}^1 - \alpha\nabla_\phi\hat{g}_T^2(\widetilde{\phi}_{MK}^1)) - g_T^3(\phi_*^3)$$
$$\leq M_{g_T}\|\widetilde{\phi}_{MK}^1 - \widetilde{\phi}_{M*}^1\| + M_{g_T}\|\widetilde{\phi}_*^1 - \phi_*^1\| + M_{g_T}\alpha\|\nabla_\phi\hat{g}_T^2(\widetilde{\phi}_{MK}^1) - \nabla_\phi\hat{g}_T^1(\widetilde{\phi}_{M*}^1)\| + M_{g_T}\delta_{13}.$$

*For L2O Transfer Learning, the generalization error is shown as below:*

$$g_T^3(\widetilde{\phi}_K^1 - \alpha\nabla_\phi\hat{g}_T^2(\phi)) - g_T^3(\phi_*^3)$$
$$\leq M_{g_T}\|\widetilde{\phi}_K^1 - \widetilde{\phi}_{M*}^1\| + M_{g_T}\|\widetilde{\phi}_*^1 - \phi_*^1\| + M_{g_T}\alpha\|\nabla_\phi\hat{g}_T^2(\widetilde{\phi}_K^1) - \nabla_\phi\hat{g}_T^1(\widetilde{\phi}_{M*}^1)\| + M_{g_T}\delta_{13},$$

*where $\delta_{13} = \|\phi_*^1 - \phi_*^3\|$ and $\widetilde{\phi}_K^1$ represents transfer learning L2O learned point after $K$ epochs.*

The generalization error gap between M-L2O and Trasnfer Learning can be categorized into two parts: (i) Difference between $\|\widetilde{\phi}_{MK}^1 - \widetilde{\phi}_{M*}^1\|$ and $\|\widetilde{\phi}_K^1 - \widetilde{\phi}_{M*}^1\|$; (ii) Difference between $\|\nabla_\phi\hat{g}_T^2(\widetilde{\phi}_{MK}^1) - \nabla_\phi\hat{g}_T^1(\widetilde{\phi}_{M*}^1)\|$ and $\|\nabla_\phi\hat{g}_T^2(\widetilde{\phi}_K^1) - \nabla_\phi\hat{g}_T^1(\widetilde{\phi}_{M*}^1)\|$. If we assume $\|\nabla_\phi\hat{g}_T^2(\phi)\| \approx \|\nabla_\phi\hat{g}_T^1(\phi)\|$, then both differences can be characterized by the gap between $\|\widetilde{\phi}_{MK}^1 - \widetilde{\phi}_{M*}^1\|$ and $\|\widetilde{\phi}_K^1 - \widetilde{\phi}_{M*}^1\|$. Since $\widetilde{\phi}_{MK}^1$ is trained to converge to $\widetilde{\phi}_{M*}^1$ as $K$ increases, it is natural to see that M-L2O ($\|\widetilde{\phi}_{MK}^1 - \widetilde{\phi}_{M*}^1\|$) enjoys smaller generalization error compared to Transfer Learning ( $\|\widetilde{\phi}_K^1 - \widetilde{\phi}_{M*}^1\|$).

We further distinguish our theory from previous theoretical works. In the L2O area, Heaton et al. (2020) analyzed the convergence of proposed safe-L2O but not generalization while Chen et al. (2020c) analyzed the generalization of quadratic-based L2O. Instead, we develop the generalization on a general class of L2O problems. In the meta-learning area, the previous works have demonstrated the convergence and generalization of MAML (Ji et al., 2022; Fallah et al., 2021). Instead, we leverage the MAML results in L2O domain to measure the learned point and training optimal point distance. Then, our L2O theory further characterizes transferability of learned point on meta testing tasks. Overall, our developed theorem is based on both L2O and meta learning results. In conclusion, our theoretical novelty lies in three aspects: ① Rigorously characterizing a generic class of L2O generalization. ② Incorporating the MAML results in our meta-learning analysis. ③ Theoretically proving that both training-like and testing-like adaptation contribute to better generalization in L2O.

## 5 EXPERIMENTS

In this section, we provide a comprehensive description of the experimental settings and present the results we obtained. Our findings demonstrate a high degree of consistency between the empirical observations and the theoretical outcomes.

### 5.1 EXPERIMENTAL CONFIGURATIONS

**Backbones and observations.** For all our experiments, we use a single-layer LSTM network with 20 hidden units as the backbone. We adopt the methodology proposed by Lv et al. (2017) and Chen et al. (2020a) to utilize the parameters' gradients and their corresponding normalized momentum to construct the observation vectors.

**Optimizees.** We conduct experiments on three distinct optimizees, namely LASSO, Quadratic, and Rosenbrock (Rosenbrock, 1960). The formulation of the Quadratic problem is $\min_{\boldsymbol{x}} \frac{1}{2}\|\mathbf{A}\boldsymbol{x} - \mathbf{b}\|_2$ and the formulation of the LASSO problem is $\min_{\boldsymbol{x}} \frac{1}{2}\|\mathbf{A}\boldsymbol{x} - \mathbf{b}\|_2 + \lambda\|\boldsymbol{x}\|_1$, where $\mathbf{A} \in \mathbb{R}^{d \times d}$, $\mathbf{b} \in \mathbb{R}^d$. We set $\lambda = 0.005$. The precise formulation of the Rosenbrock problem is available in Section A.6. During the meta-training and testing stage, the optimizees $\xi_{\text{train}}$ and $\xi_{\text{test}}$ are drawn from the pre-specified distributions $D_{\text{train}}$ and $D_{\text{test}}$, respectively. Similarly, the optimizees $\xi_{\text{adapt}}$ used during adaptation are sampled from the distribution $D_{\text{adapt}}$.

**Baselines and Training Settings.** We compare M-L2O against three baselines: (1) *Vanilla L2O*, where we train a randomly initialized L2O optimizer on $\xi_{\text{adapt}}$ for only 5 steps; (2) *Transfer Learning* (TL), where we first meta-train a randomly initialized L2O optimizer on $\xi_{\text{train}}$, and then fine-tune on $\xi_{\text{adapt}}$ for 5 steps; (3) *Direct Transfer* (DT), where we meta-train a randomly initialized L2O optimizer on $\xi_{\text{train}}$ only. M-L2O adopts a fair experimental setting, whereby we meta-train on $\xi_{\text{train}}$ and adapt on the same $\xi_{\text{adapt}}$. We evaluate these methods using the same set of optimizees for testing (*i.e.*, $\xi_{\text{test}}$), and report the minimum ***logarithmic*** value of the objective functions achieved for these optimizees.

For all experiments, we set the number of optimizee iterations, denoted by $T$, to 20 when meta-training the L2O optimizers and adapting to optimizees. Notably, in large scale experiments involving neural networks as tasks, the common choice for $T$ is 5 (Zheng et al., 2022; Shen et al., 2021). However, in our experiments, we set $T$ to 20 to achieve better experimental performance. The value of the total epochs, denoted by $K$, is set to 5000, and we adopt the curriculum learning technique (Chen et al., 2020a) to dynamically adjust the number of epochs per task, denoted by $S$. To update the weights of the optimizers ($\phi$), we use Adam (Kingma & Ba, 2014) with a fixed learning rate of $1 \times 10^{-4}$.

### 5.2 FAST ADAPTATION RESULTS OF M-L2O

**Experiments on LASSO optimizees.** We begin with experiments on LASSO optimizees. Specifically, for $\xi_{\text{train}}$, the coefficient matrix $\mathbf{A}$ is generated by sampling from a mixture of uniform distributions comprising $\{\text{U}(0, 0.1), \text{U}(0, 0.5), \text{U}(0, 1)\}$. In contrast, for $\xi_{\text{test}}$ and $\xi_{\text{adapt}}$, the coefficient matrices $\mathbf{A}$ are obtained by sampling from a normal distribution with a standard deviation of $\sigma$. We conduct experiments with $\sigma = 50$ and $\sigma = 100$, and report the results in Figures 2a and 2b. Our findings demonstrate that:

① The Vanilla L2O approach, which relies on only five steps of adaptation from initialization on $\xi_{\text{adapt}}$, exhibits the weakest performance, as evidenced by the largest values of the objective function.

② Although Direct Transfer (DT) is capable of learning optimization rules from the training optimizees, the larger variance in coefficients among the testing optimizees renders the learned rules inadequate for generalization.

③ The superiority of Transfer Learning (TL) over DT highlights the values of adaptation when the testing optimizees deviates significantly from those seen in training, as the optimizer is presumably able to acquire new knowledge during the adaptation process.

④ Finally, M-L2O exhibits consistent and notably faster convergence speed compared to other baseline methods. Moreover, it demonstrates the best performance overall, reducing the logarithm of the objective values by approximately 0.2 and 1 when $\sigma = 50$ and $\sigma = 100$, respectively. M-L2O's superior performance can be attributed to its ability to learn well-adapted initial weights for optimizers, which enables rapid self-adaptation, thus leading to better performance in comparison to the baseline methods.

Table 1: Minimum logarithm loss of different methods on LASSO at different levels of $\sigma$. We report the 95% confidence interval from 10 repeated runs.

| $\sigma$ | Methods | | | |
| --- | --- | --- | --- | --- |
| | Vanilla L2O | M-L2O | DT | TL |
| 10 | $0.033_{\pm 0.661}$ | $-3.712_{\pm 0.004}$ | $\mathbf{-4.233}_{\pm 0.016}$ | $-4.077_{\pm 0.015}$ |
| 25 | $1.559_{\pm 0.789}$ | $-3.433_{\pm 0.011}$ | $\mathbf{-4.125}_{\pm 0.011}$ | $-4.019_{\pm 0.017}$ |
| 50 | $0.550_{\pm 0.476}$ | $\mathbf{-3.285}_{\pm 0.013}$ | $-3.098_{\pm 0.021}$ | $-3.181_{\pm 0.011}$ |
| 100 | $2.435_{\pm 1.500}$ | $\mathbf{-2.408}_{\pm 0.037}$ | $-1.775_{\pm 0.034}$ | $-1.961_{\pm 0.050}$ |
| 200 | $4.104_{\pm 1.300}$ | $\mathbf{-1.396}_{\pm 0.035}$ | $-0.453_{\pm 0.075}$ | $-0.982_{\pm 0.086}$ |

**Experiments on Quadratic optimizees.** We continue to assess the performance of our approach on a different optimizee, *i.e.*, the Quadratic problem. The coefficient matrices $\mathbf{A}$ of the optimizees are also randomly sampled from a normal distribution. We conduct two evaluations, with $\sigma$ values of 50 and 100, respectively, and present the outcomes in Figure 3. Notably, the results show a similar trend to those we obtained in the previous experiments.

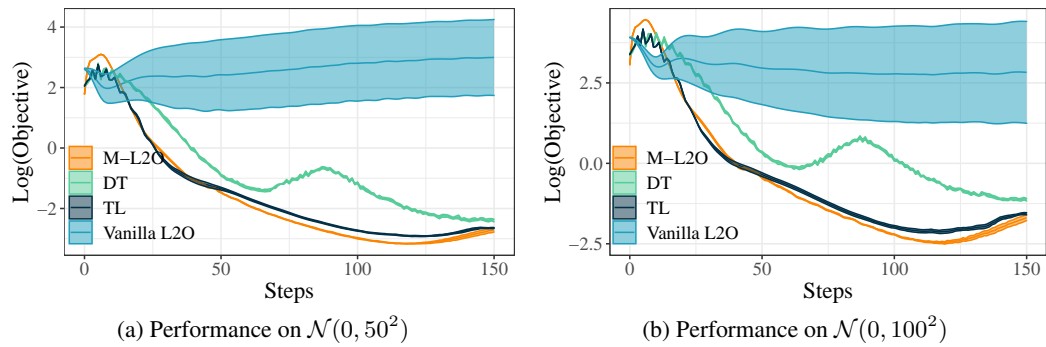

(a) Performance on $\mathcal{N}(0, 50^2)$      (b) Performance on $\mathcal{N}(0, 100^2)$

Figure 2: Comparison of convergence speeds on target distribution of LASSO optimizees. We repeat the experiments for 10 times and show the 95% confidence intervals in the figures.

**More LASSO experiments.** We proceed to investigate the impact of varying the standard deviation $\sigma$ of the distributions we used to sample the coefficient matrices $\mathbf{A}$ for $\xi_{\text{adapt}}$ and $\xi_{\text{test}}$. The minimum logarithm of the objective value for each method is reported in Table 1. Our findings reveal that:

① At lower levels of $\sigma$, it is not always necessary, and may even be unintentionally harmful, to use adaptation for normally trained L2O. Although M-L2O produces satisfactory results, it exhibits slightly lower performance than TL, which could be due to the high similarity between the training and testing tasks. Since M-L2O's objective is to identify optimal general initial points, L2O optimizers trained directly on related and similar tasks may effectively generalize. However, after undergoing adaptation on a single task, L2O optimizers may discard certain knowledge acquired during meta-training that could be useful for novel but similar tasks.

② Nevertheless, as the degree of similarity between training and testing tasks is declines, as characterized by an increasing value of $\sigma$, M-L2O begins to demonstrate considerable advantage. For values of $\sigma$ greater than 50, M-L2O exhibits consistent performance advantages that exceed $0.1$ in terms of logarithmic loss. This observation empirically supports that the learned initial weights facilitate rapid adaptation to new tasks that are "out-of-distribution", and that manifest large deviations.

## 5.3 ADAPTATION WITH SAMPLES FROM DIFFERENT TASK DISTRIBUTION

In Section 5.2, we impose a constraint on the standard deviation of the distribution used to sample $\mathbf{A}$, ensuring that is identical for both the optimizees for adaptation and testing. However, it is noteworthy that this constraint is not mandatory, given that our theory can accommodate adaptation and testing optimizees with different distributions. Consequently, we conduct an experiment on LASSO optimizees with varying standard deviations of the distribution, from which the matrices $\mathbf{A}$ for optimizee $\xi_{\text{adapt}}$ is drawn. Specifically, we sample $\sigma$ with smaller values that more resemble the training tasks, as well as larger values that are more similar to the testing task ($\sigma = 100$).

In Theorem 1, we have characterized the generalization of M-L2O with flexible distribution adaptation tasks. The theoretical analysis suggests that a similar geometry landscape (smaller

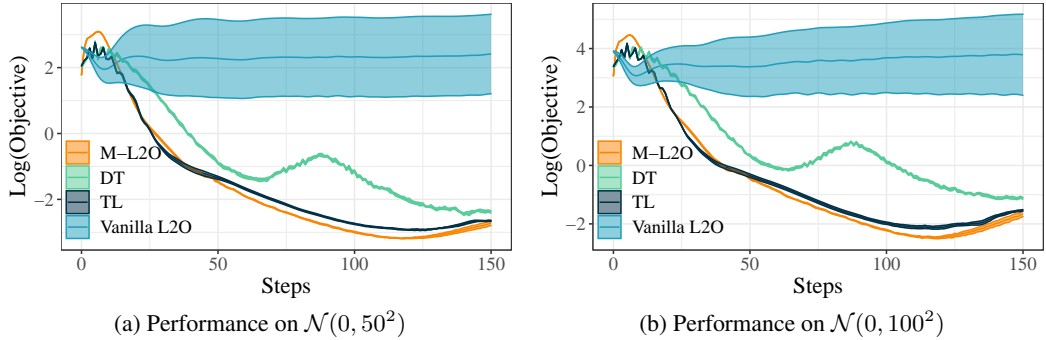

(a) Performance on $\mathcal{N}(0, 50^2)$           (b) Performance on $\mathcal{N}(0, 100^2)$

Figure 3: Comparison of convergence speeds on target distribution of Quadratic optimizees. We repeat the experiments for 10 times and show the 95% confidence intervals are shown in the figures.

$\Delta_{12}, \widetilde{\Delta}_{12}$) between the training and adaptation tasks, can lead to a reduction in generalization loss as defined in Equation (7). This claim has been corroborated by the results of our experiments,

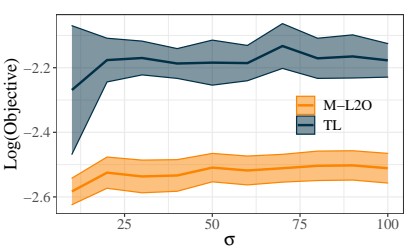

Figure 4: Performance on LASSO optimizees. We vary the standard deviation of the distribution used for sampling the weight matrix $\mathbf{A}$ for adaptation optimizees. We visualize both the mean and the confidence interval in the figure.

as presented in Figure 4. When the $\sigma$ is similar to the training tasks (*e.g.,* 10), implying a smaller $\Delta 12$ and $\widetilde{\Delta}_{12}$, M-L2O demonstrates superior testing performance. In conclusion, incorporating training-like adaptation tasks can lead to better generalization performance.

Meanwhile, it is reasonable to suggest that the task differences between adaptation and testing, denoted by $(\Delta_{23}, \widetilde{\Delta}_{23})$, may also have an impact on M-L2O's generalization ability. Intuitively, if the optimizer is required to adapt to testing optimizees, the adapted optimize should demonstrate strong generalization ability on other optimizees that are similar. In order to have a deeper understanding of the relationship between the generalization ability and the difference between adaptation and testing tasks, we rewrite M-L2O generalization error in Theorem 1 in the following form with $\widetilde{\phi}^3_* = \widetilde{\phi}^3_{M*} - \alpha \nabla_\phi \hat{g}^3_T(\widetilde{\phi}^3_{M*})$ and $\widetilde{\delta}^M_{13} = \|\widetilde{\phi}^1_{M*} - \widetilde{\phi}^3_{M*}\|$:

$$g^3_T(\widetilde{\phi}^1_{MK} - \alpha \nabla_\phi \hat{g}^2_T(\widetilde{\phi}^1_{MK})) - g^3_T(\phi^3_*)$$
$$\leq M_{g_T}(\|\widetilde{\phi}^1_{MK} - \widetilde{\phi}^1_{M*}\| + \|\widetilde{\phi}^3_* - \phi^3_*\| + \alpha \|\nabla_\phi \hat{g}^2_T(\widetilde{\phi}^1_{MK}) - \nabla_\phi \hat{g}^3_T(\widetilde{\phi}^3_{M*})\| + \widetilde{\delta}^M_{13}). \qquad (8)$$

In Equation (8), M-L2O generalization error is partly captured by $\|\nabla_\phi \hat{g}^2_T(\widetilde{\phi}^1_{MK}) - \nabla_\phi \hat{g}^3_T(\widetilde{\phi}^3_{M*})\|$ which is controlled by difference in optimizers (*i.e.,* $\|\nabla_\phi \hat{g}^2_T(\phi) - \nabla_\phi \hat{g}^3_T(\phi)\|$). From Proposition 1, we know that this term is determined by difference in optimizees, denoted by $\Delta_{23}$ and $\widetilde{\Delta}_{23}$. Similar to the results established in Theorem 1, we can deduce that superior testing performance is connected with a smaller difference between testing and adaptation optimizees. This result has been demonstrated in Figure 4 where TL generalizes well with larger $\sigma$ (more testing-like). Moreover, M-L2O also benefits from larger $\sigma$ values (e.g., $\sigma = 100$) in certain scenarios.

To summarize, both training-like and testing-like adaptation task can lead to improved testing performance. As shown in Figure 4, training-like adaptation results in better generalization in L2O. One possible explanation is that when the testing task significantly deviates from the training tasks, it becomes highly challenging for the optimizer to generalize well within limited adaptation steps. In such scenarios, the training-like adaptation provides a more practical solution.

## 6   CONCLUSION AND DISCUSSION

In this paper, we propose a self-adapted L2O algorithm (M-L2O), which is incorporated with meta adaptation. Such a design enables the optimizer to reach a well-adapted initial point, facilitating its adaptation ability with only a few updates. Our superior generalization performances in out-of-distribution tasks have been theoretically characterized and empirically validated across various scenarios. Furthermore, the comprehensive empirical results demonstrate that training-like adaptation tasks can contribute to better testing generalization, which is consistent with our theoretical analysis. One potential future direction is to develop a convergence analysis for L2O. It will be more interesting to consider meta adaptation in analyzing L2O convergence from a theoretical view.

## ACKNOWLEDGEMENTS

The work of Y. Liang was supported in part by the U.S. National Science Foundation under the grants ECCS-2113860 and DMS-2134145. The work of Z. Wang was supported in part by the U.S. National Science Foundation under the grant ECCS-2145346 (CAREER Award).

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

# A  APPENDIX

## A.1  RESTATEMENT OF ASSUMPTION 3

**Assumption 5** (Restatement of Assumption 3). *We assume Lipschitz properties for all functions $l_i(\theta)(i = 1, 2, 3)$ as follows:*

a) *$l_i(\theta)$ is $M$-Lipschitz, i.e., for any $\theta_1$ and $\theta_2$, $\|l_i(\theta_1) - l_i(\theta_2)\| \leq M\|\theta_1 - \theta_2\|(i = 1, 2, 3)$.*

b) *$\nabla l_i(\theta)$ is $L$-Lipschitz, i.e., for any $\theta_1$ and $\theta_2$, $\|\nabla l_i(\theta_1) - \nabla l_i(\theta_2)\| \leq L\|\theta_1 - \theta_2\|(i = 1, 2, 3)$.*

c) *$\nabla^2 l_i(\theta)$ is $\rho$-Lipschitz, i.e., for any $\theta_1$ and $\theta_2$, $\|\nabla^2 l_i(\theta_1) - \nabla^2 l_i(\theta_2)\| \leq \rho\|\theta_1 - \theta_2\|(i = 1, 2, 3)$.*

d) *$m(z, \phi)$ is $M_{m1}$-Lipschitz w.r.t. $z$ and $M_{m2}$-Lipschitz w.r.t. $\phi$, i.e.,*

$$\begin{aligned}\|m(z_1, \phi) - m(z_2, \phi)\| &\leq M_{m1}\|z_1 - z_2\| \quad \text{for any } z_1 \text{ and } z_2,\\ \|m(z, \phi_1) - m(z, \phi_2)\| &\leq M_{m2}\|\phi_1 - \phi_2\| \quad \text{for any } \phi_1 \text{ and } \phi_2.\end{aligned}$$

e) *$\nabla_\phi^2 \theta_T(\phi)$ is $\rho_\theta$-Lipschitz, i.e., for any $\phi_1$ and $\phi_2$, $\|\nabla_\phi^2 \theta_T(\phi_1) - \nabla_\phi^2 \theta_T(\phi_2)\| \leq \rho_\theta\|\phi_1 - \phi_2\|$.*

*The above Assumptions (a)(b)(c) also hold for stochastic $\hat{l}_i(\theta)$, $\nabla\hat{l}_i(\theta)$ and $\nabla_\theta^2\hat{l}_i(\theta)(i = 1, 2, 3)$.*

## A.2  PROOF OF SUPPORTING LEMMAS (LEMMA 12 CORRESPONDS TO PROPOSITION 1)

**Lemma 1.** *Based on update procedure of $\theta_t(\theta)$, we obtain*

$$\nabla_\phi\theta_T(\phi) = \sum_{i=0}^{T-1}\left(\prod_{j=i+1}^{T-1}\left(I + \nabla_1 m(\nabla_\theta\hat{l}(\theta_{T+i-j}), \phi)\nabla_\theta^2\hat{l}(\theta_{T+i-j})\right)\nabla_2 m(\nabla_\theta\hat{l}(\theta_i), \phi)\right).$$

*Proof.* The $\theta_t(\phi)$ update process is shown below:

$$\theta_{t+1}(\phi) = \theta_t(\phi) + m(z_t, \phi).$$

If we only consider $z_t(\theta_t; \zeta_t) = \nabla_\theta l(\theta_t(\phi); \zeta_t) = \nabla_\theta\hat{l}(\theta_t)$, then we obtain

$$\begin{aligned}\nabla_\phi\theta_{t+1}(\phi) &= \nabla_\phi\theta_t(\phi) + \nabla_\phi m(\nabla_\theta\hat{l}(\theta_t), \phi)\\ &= \nabla_\phi\theta_t(\phi) + \nabla_1 m(\nabla_\theta\hat{l}(\theta_t), \phi)\nabla_\theta^2\hat{l}(\theta_t)\nabla_\phi\theta_t(\phi) + \nabla_2 m(\nabla_\theta\hat{l}(\theta_t), \phi)\\ &= (I + \nabla_1 m(\nabla_\theta\hat{l}(\theta_t), \phi)\nabla_\theta^2\hat{l}(\theta_t))\nabla_\phi\theta_t(\phi) + \nabla_2 m(\nabla_\theta\hat{l}(\theta_t), \phi).\end{aligned}$$

If we iterate the above equation from $t = 0$ to $T$, then we obtain

$$\begin{aligned}\nabla_\phi\theta_T(\phi) = &\sum_{i=0}^{T-1}\left(\prod_{j=i+1}^{T-1}\left(I + \nabla_1 m(\nabla_\theta\hat{l}(\theta_{T+i-j}), \phi)\nabla_\theta^2\hat{l}(\theta_{T+i-j})\right)\nabla_2 m(\nabla_\theta\hat{l}(\theta_i), \phi)\right)\\ &+ \prod_{i=1}^{T}\left(I + \nabla_1 m(\nabla_\theta\hat{l}(\theta_{T-i}), \phi)\nabla_\theta^2\hat{l}(\theta_{T-i})\right)\nabla_\phi\theta_0,\end{aligned}$$

We assume $\theta_0$ is randomly sampled and independent from $\phi$, then we obtain

$$\nabla_\phi\theta_T(\phi) = \sum_{i=0}^{T-1}\left(\prod_{j=i+1}^{T-1}\left(I + \nabla_1 m(\nabla_\theta\hat{l}(\theta_{T+i-j}), \phi)\nabla_\theta^2\hat{l}(\theta_{T+i-j})\right)\nabla_2 m(\nabla_\theta\hat{l}(\theta_i), \phi)\right).$$

$\square$

**Lemma 2.** *If we assume that $\theta_0(\phi_1) = \theta_0(\phi_2)$, based on Assumption 3, then we obtain*

$$\|\theta_T(\phi_1) - \theta_T(\phi_2)\| \leq \left(((M_{m1}L + 1)^{T-1} - 1)\frac{M_{m2}}{M_{m1}L}\right)\|\phi_1 - \phi_2\| = M_{\theta T}\|\phi_1 - \phi_2\|. \quad (9)$$

*Proof.* Based on the iterate procedure of $\theta_T(\phi)$, we obtain

$$
\|\theta_T(\phi_1) - \theta_T(\phi_2)\|
$$
$$
\overset{(i)}{=} \left\| \sum_{t=1}^{T-1} (m(\nabla_\theta \hat{l}(\theta_t(\phi_1)), \phi_1) - m(\nabla_\theta \hat{l}(\theta_t(\phi_2)), \phi_2)) \right\|
$$
$$
= \left\| \sum_{t=1}^{T-1} (m(\nabla_\theta \hat{l}(\theta_t(\phi_1)), \phi_1) - m(\nabla_\theta \hat{l}(\theta_t(\phi_1)), \phi_2) + m(\nabla_\theta \hat{l}(\theta_t(\phi_1)), \phi_2) \right.
$$
$$
\left. - m(\nabla_\theta \hat{l}(\theta_t(\phi_2)), \phi_2)) \right\|
$$
$$
\leq \left\| \sum_{t=1}^{T-1} (m(\nabla_\theta \hat{l}(\theta_t(\phi_1)), \phi_1) - m(\nabla_\theta \hat{l}(\theta_t(\phi_1)), \phi_2)) \right\|
$$
$$
+ \left\| \sum_{t=1}^{T-1} m(\nabla_\theta \hat{l}(\theta_t(\phi_1)), \phi_2) - m(\nabla_\theta \hat{l}(\theta_t(\phi_2)), \phi_2)) \right\|
$$
$$
\overset{(ii)}{\leq} \left\| \sum_{t=1}^{T-1} M_{m2} \|\phi_1 - \phi_2\| \right\| + \left\| \sum_{t=1}^{T-1} M_{m1} \|\nabla_\theta \hat{l}(\theta_t(\phi_1)) - \nabla_\theta \hat{l}(\theta_t(\phi_2))\| \right\|
$$
$$
\leq (T-1) M_{m2} \|\phi_1 - \phi_2\| + M_{m1} \sum_{t=1}^{T-1} \|\nabla_\theta \hat{l}(\theta_t(\phi_1)) - \nabla_\theta \hat{l}(\theta_t(\phi_2))\|
$$
$$
\overset{(iii)}{\leq} (T-1) M_{m2} \|\phi_1 - \phi_2\| + M_{m1} L \sum_{t=1}^{T-1} \|\theta_t(\phi_1) - \theta_t(\phi_2)\|,
$$

where $(i)$ follows from Equation (1), $(ii)$ and $(iii)$ from Assumption 3. If we further iterate it from $t = 0$ to $T$, we obtain

$$
\|\theta_T(\phi_1) - \theta_T(\phi_2)\| \leq \left( ((M_{m1}L+1)^{T-1} - 1) \frac{M_{m2}}{M_{m1}L} \right) \|\phi_1 - \phi_2\| = M_{\theta T} \|\phi_1 - \phi_2\|.
$$

$\square$

**Lemma 3.** *If we define $A_i(\phi) = \nabla_2 m(\nabla_\theta \hat{l}(\theta_i(\phi_1)), \phi_1)$, based on Assumption 3 and Lemma 2, we obtain*

$$
\|A_i(\phi_1) - A_i(\phi_2)\| \leq M_{Ai} \|\phi_1 - \phi_2\|,
$$

*where $M_{Ai} = L_{m2} + L_{m1} L M_{\theta i}$.*

*Proof.* Based on the definition of $A_i(\phi)$, we have

$$
\|A_i(\phi_1) - A_i(\phi_2)\|
$$
$$
= \|\nabla_2 m(\nabla_\theta \hat{l}(\theta_i(\phi_1)), \phi_1) - \nabla_2 m(\nabla_\theta \hat{l}(\theta_i(\phi_2)), \phi_2)\|
$$
$$
= \|\nabla_2 m(\nabla_\theta \hat{l}(\theta_i(\phi_1)), \phi_1) - \nabla_2 m(\nabla_\theta \hat{l}(\theta_i(\phi_1)), \phi_2)
$$
$$
+ \nabla_2 m(\nabla_\theta \hat{l}(\theta_i(\phi_1)), \phi_2) - \nabla_2 m(\nabla_\theta \hat{l}(\theta_i(\phi_2)), \phi_2)\|
$$
$$
\overset{(i)}{\leq} L_{m2} \|\phi_1 - \phi_2\| + L_{m1} \|\nabla_\theta \hat{l}(\theta_i(\phi_1)) - \nabla_\theta \hat{l}(\theta_i(\phi_2))\|
$$
$$
\leq L_{m2} \|\phi_1 - \phi_2\| + L_{m1} L \|\theta_i(\phi_1) - \theta_i(\phi_2)\|
$$
$$
\overset{(ii)}{\leq} (L_{m2} + L_{m1} L M_{\theta i}) \|\phi_1 - \phi_2\| = M_{Ai} \|\phi_1 - \phi_2\|,
$$

where $(i)$ follows from Assumption 3, $(ii)$ follows from Lemma 2. $\square$

**Lemma 4.** *We first define $B_i(\phi) = \nabla_1 m(\nabla_\theta \hat{l}(\theta_i(\phi)), \phi) \nabla_\theta^2 \hat{l}(\theta_i(\phi))$. Based on the Lemma 2 and Assumption 3, we obtain*

$$
\|B_i(\phi_1) - B_i(\phi_2)\| \leq M_{Bi} \|\phi_1 - \phi_2\|,
$$

*where $M_{Bi} = M_{m1} \rho M_{\theta i} + L L_{m2} + L^2 L_{m1} M_{\theta i}$.*

*Proof.* Based on the definition of $B_i(\phi)$, we have

$\|B_i(\phi_1) - B_i(\phi_2)\|$

$= \|\nabla_1 m(\nabla_\theta \hat{l}(\theta_i(\phi_1)), \phi_1)\nabla_\theta^2 \hat{l}(\theta_i(\phi_1)) - \nabla_1 m(\nabla_\theta \hat{l}(\theta_i(\phi_2)), \phi_2)\nabla_\theta^2 \hat{l}(\theta_i(\phi_2))\|$

$= \|\nabla_1 m(\nabla_\theta \hat{l}(\theta_i(\phi_1)), \phi_1)\nabla_\theta^2 \hat{l}(\theta_i(\phi_1)) - \nabla_1 m(\nabla_\theta \hat{l}(\theta_i(\phi_1)), \phi_1)\nabla_\theta^2 \hat{l}(\theta_i(\phi_2))$

$\quad + \nabla_1 m(\nabla_\theta \hat{l}(\theta_i(\phi_1)), \phi_1)\nabla_\theta^2 \hat{l}(\theta_i(\phi_2)) - \nabla_1 m(\nabla_\theta \hat{l}(\theta_i(\phi_2)), \phi_2)\nabla_\theta^2 \hat{l}(\theta_i(\phi_2))\|$

$\overset{(i)}{\leq} M_{m1}\|\nabla_\theta^2 \hat{l}(\theta_i(\phi_1)) - \nabla_\theta^2 \hat{l}(\theta_i(\phi_2))\| + L\|\nabla_1 m(\nabla_\theta \hat{l}(\theta_i(\phi_1)), \phi_1) - \nabla_1 m(\nabla_\theta \hat{l}(\theta_i(\phi_2)), \phi_2)\|$

$\overset{(ii)}{\leq} M_{m1}\rho M_{\theta i}\|\phi_1 - \phi_2\| + L\|\nabla_1 m(\nabla_\theta \hat{l}(\theta_i(\phi_1)), \phi_1) - \nabla_1 m(\nabla_\theta \hat{l}(\theta_i(\phi_1)), \phi_2)$

$\quad + \nabla_1 m(\nabla_\theta \hat{l}(\theta_i(\phi_1)), \phi_2) - \nabla_1 m(\nabla_\theta \hat{l}(\theta_i(\phi_2)), \phi_2)\|$

$\overset{(iii)}{\leq} M_{m1}\rho M_{\theta i}\|\phi_1 - \phi_2\| + LL_{m2}\|\phi_1 - \phi_2\| + LL_{m1}\|\nabla_\theta \hat{l}(\theta_i(\phi_1)) - \nabla_\theta \hat{l}(\theta_i(\phi_2))\|$

$\leq (M_{m1}\rho M_{\theta i} + LL_{m2} + L^2 L_{m1} M_{\theta i})\|\phi_1 - \phi_2\| = M_{Bi}\|\phi_1 - \phi_2\|,$

where $(i)$ and $(iii)$ follows from Assumption 3, $(ii)$ follows from Lemma 2. $\qquad\square$

**Lemma 5.** *Based on Assumption 3 and Lemmas 1, 3 and 4, then we obtain*

$$\|\nabla_\phi \theta_T(\phi_1) - \nabla_\phi \theta_T(\phi_2)\| \leq L_{\theta T}\|\phi_1 - \phi_2\|, \tag{10}$$

*where* $L_{\theta T} = \sum_{i=0}^{T-1}(1 + M_{m1}L)^{T-i-1}M_{Ai} + \sum_{i=0}^{T-1} M_{m2}(1 + M_{m1}L)^{T-i-2}\sum_{j=i+1}^{T-1} M_{B(T+i-j)}.$

*Proof.* Based on the definition of $\nabla_\phi \theta_T(\phi)$ in Lemma 1, we obtain

$\|\nabla_\phi \theta_T(\phi_1) - \nabla_\phi \theta_T(\phi_2)\|$

$\overset{(i)}{\leq} \sum_{i=0}^{T-1}\Bigg\|\prod_{j=i+1}^{T-1}\Bigg(I + \nabla_1 m(\nabla_\theta \hat{l}(\theta_{T+i-j}(\phi_1)), \phi_1)\nabla_\theta^2 \hat{l}(\theta_{T+i-j}(\phi_1))\Bigg)\nabla_2 m(\nabla_\theta \hat{l}(\theta_i(\phi_1)), \phi_1)$

$\quad - \prod_{j=i+1}^{T-1}\Bigg(I + \nabla_1 m(\nabla_\theta \hat{l}(\theta_{T+i-j}(\phi_2)), \phi_2)\nabla_\theta^2 \hat{l}(\theta_{T+i-j}(\phi_2))\Bigg)\nabla_2 m(\nabla_\theta \hat{l}(\theta_i(\phi_2)), \phi_2)\Bigg\|$

$\overset{(ii)}{=} \sum_{i=0}^{T-1}\Bigg\|\prod_{j=i+1}^{T-1}\Bigg(I + B_{T+i-j}(\phi_1)\Bigg)A_i(\phi_1) - \prod_{j=i+1}^{T-1}\Bigg(I + B_{T+i-j}(\phi_2)\Bigg)A_i(\phi_2)\Bigg\|$

$= \sum_{i=0}^{T-1}\Bigg\|\prod_{j=i+1}^{T-1}\Bigg(I + B_{T+i-j}(\phi_1)\Bigg)A_i(\phi_1) - \prod_{j=i+1}^{T-1}\Bigg(I + B_{T+i-j}(\phi_1)\Bigg)A_i(\phi_2)$

$\quad + \prod_{j=i+1}^{T-1}\Bigg(I + B_{T+i-j}(\phi_1)\Bigg)A_i(\phi_2) - \prod_{j=i+1}^{T-1}\Bigg(I + B_{T+i-j}(\phi_2)\Bigg)A_i(\phi_2)\Bigg\|$

$\overset{(iii)}{\leq} \sum_{i=0}^{T-1}\Bigg((1 + M_{m1}L)^{T-i-1}\|A_i(\phi_1) - A_i(\phi_2)\| + M_{m2}\Bigg\|\prod_{j=i+1}^{T-1}\Big(I + B_{T+i-j}(\phi_1)\Big)$

$\quad - \prod_{j=i+1}^{T-1}\Big(I + B_{T+i-j}(\phi_2)\Big)\Bigg\|\Bigg)$

$\leq \sum_{i=0}^{T-1}\Bigg((1 + M_{m1}L)^{T-i-1}\|A_i(\phi_1) - A_i(\phi_2)\| + M_{m2}(1 + M_{m1}L)^{T-i-2}$

$\quad \sum_{j=i+1}^{T-1}\|B_{T+i-j}(\phi_1) - B_{T+i-j}(\phi_2)\|\Bigg)$

$\overset{(iv)}{\leq} \sum_{i=0}^{T-1}\Bigg((1 + M_{m1}L)^{T-i-1}M_{Ai}\|\phi_1 - \phi_2\| + M_{m2}(1 + M_{m1}L)^{T-i-2}$

$$\sum_{j=i+1}^{T-1} M_{B(T+i-j)}\|\phi_1 - \phi_2\| \Bigg)$$

$$= \Bigg( \sum_{i=0}^{T-1}(1 + M_{m1}L)^{T-i-1}M_{Ai} + \sum_{i=0}^{T-1} M_{m2}(1 + M_{m1}L)^{T-i-2}\sum_{j=i+1}^{T-1} M_{B(T+i-j)} \Bigg)\|\phi_1 - \phi_2\|$$

$$= L_{\theta T}\|\phi_1 - \phi_2\|,$$

where $(i)$ is based on Lemma 1, $(ii)$ is based on the fact that $A_i(\phi_1) = \nabla_2 m(\nabla_\theta \hat{l}(\theta_i(\phi_1)), \phi_1)$, $B_{T+i-j}(\phi_1) = \nabla_1 m(\nabla_\theta \hat{l}(\theta_{T+i-j}(\phi_1)), \phi_1)\nabla_\theta^2 \hat{l}(\theta_{T+i-j}(\phi_1))$, $(iii)$ follows from Assumption 3 and $(iv)$ follows from Lemma 3 and 4. $\qquad\square$

**Lemma 6.** *Based on Lemmas 2, 5 and Assumption 3, we obtain*

$$\|\nabla_\phi \hat{g}_T(\phi_1) - \nabla_\phi \hat{g}_T(\phi_2)\| \le L_{g_T}\|\phi_1 - \phi_2\|,$$

*where $L_{g_T} = ML_{\theta T} + LM_{\theta T}^2$, $L_{\theta T}$ is defined in Lemma 5, $M_{\theta T}$ is defined in Lemma 2.*

*Proof.* We assume all functions share the same starting point $\theta_0$, then we have

$$\|\nabla_\phi \hat{g}_T(\phi_1) - \nabla_\phi \hat{g}_T(\phi_2)\|$$
$$= \|\nabla_\phi \hat{l}(\theta_T(\phi_1)) - \nabla_\phi \hat{l}(\theta_T(\phi_2))\|$$
$$= \|\nabla_\theta l(\theta_T(\phi_1))\nabla_\phi \theta_T(\phi_1)] - \nabla_\theta l(\theta_T(\phi_2))\nabla_\phi \theta_T(\phi_2)\|$$
$$\le \|\nabla_\theta l(\theta_T(\phi_1))\|\|\nabla_\phi \theta_T(\phi_1) - \nabla_\phi \theta_T(\phi_2)\|$$
$$\quad + \|\nabla_\theta l(\theta_T(\phi_1)) - \nabla_\theta l(\theta_T(\phi_2))\|\|\nabla_\phi \theta_T(\phi_2)\|$$
$$\overset{(i)}{\le} M\|\nabla_\phi \theta_T(\phi_1) - \nabla_\phi \theta_T(\phi_2)\| + L\|\theta_T(\phi_1) - \theta_T(\phi_2)\|\|\nabla_\phi \theta_T(\phi_2)\|$$
$$\overset{(ii)}{\le} ML_{\theta T}\|\phi_1 - \phi_2\| + LM_{\theta T}^2\|\phi_1 - \phi_2\| = (ML_{\theta T} + LM_{\theta T}^2)\|\phi_1 - \phi_2\| = L_{g_T}\|\phi_1 - \phi_2\|,$$

where $(i)$ from Assumption 3, $(ii)$ from Lemma 2 and 5. $\qquad\square$

**Lemma 7.** *Based on the Lemma 2, 5 and Assumption 3, we obtain*

$$\|\nabla_\phi^2 \hat{g}_T(\phi_1) - \nabla_\phi^2 \hat{g}_T(\phi_2)\| \le \rho_{g_T}\|\phi_1 - \phi_2\|, \tag{11}$$

*where $\rho_{g_T} = 3LM_{\theta T}L_{\theta T} + M\rho_\theta + M_{\theta T}^3\rho$.*

*Proof.* We first compute the Lipschitz condition of $\nabla_\phi \nabla_\theta \hat{l}(\theta_T(\phi))$ as follows

$$\|\nabla_\phi \nabla_\theta \hat{l}(\theta_T(\phi_1)) - \nabla_\phi \nabla_\theta \hat{l}(\theta_T(\phi_2))\|$$
$$= \|[\nabla_\phi \theta_T(\phi_1)]^T\nabla_\theta^2 \hat{l}(\theta_T(\phi_1)) - [\nabla_\phi \theta_T(\phi_2)]^T\nabla_\theta^2 \hat{l}(\theta_T(\phi_2))\|$$
$$\le \|[\nabla_\phi \theta_T(\phi_1)]^T\|\|\nabla_\theta^2 \hat{l}(\theta_T(\phi_1)) - \nabla_\theta^2 \hat{l}(\theta_T(\phi_2))\|$$
$$\quad + \|[\nabla_\phi \theta_T(\phi_1)]^T - [\nabla_\phi \theta_T(\phi_2)]^T\|\|\nabla_\theta^2 \hat{l}(\theta_T(\phi_2))\|$$
$$\overset{(i)}{\le} M_{\theta T}^2\rho\|\phi_1 - \phi_2\| + L_{\theta T}L\|\phi_1 - \phi_2\|$$
$$= (M_{\theta T}^2\rho + L_{\theta T}L)\|\phi_1 - \phi_2\|,$$

where $(i)$ follows from Lemma 2, 5 and Assumption 3. Then, based on the definition of $\nabla_\phi^2 \hat{g}_T(\phi)$, we have

$$\|\nabla_\phi^2 \hat{g}_T(\phi_1) - \nabla_\phi^2 \hat{g}_T(\phi_2)\|$$
$$= \|\nabla_\phi^2 \hat{l}(\theta_T(\phi_1)) - \nabla_\phi^2 \hat{l}(\theta_T(\phi_2))\|$$
$$= \|\nabla_\phi^2 \theta_T(\phi_1)\nabla_\theta \hat{l}(\theta_T(\phi_1)) + [\nabla_\phi^2 \theta_T^i(\phi_1)]^T\nabla_\phi \nabla_\theta \hat{l}(\theta_T(\phi_1)) - \nabla_\phi^2 \theta_T(\phi_2)\nabla_\theta \hat{l}(\theta_T(\phi_2))$$
$$\quad - [\nabla_\phi^2 \theta_T(\phi_2)]^T\nabla_\phi \nabla_\theta l_i(\theta_T(\phi_2))\|$$
$$\le \|\nabla_\phi^2 \theta_T(\phi_1)\nabla_\theta \hat{l}(\theta_T(\phi_1)) - \nabla_\phi^2 \theta_T(\phi_2)\nabla_\theta \hat{l}(\theta_T(\phi_2))\|$$

A15

$$+ \|[\nabla_\phi \theta_T(\phi_1)]^T \nabla_\phi \nabla_\theta \hat{l}(\theta_T(\phi_1)) - [\nabla_\phi \theta_T(\phi_2)]^T \nabla_\phi \nabla_\theta \hat{l}(\theta_T(\phi_2))\|$$

$$\leq \|\nabla_\phi^2 \theta_T(\phi_1)\| \|\nabla_\theta \hat{l}(\theta_T(\phi_1)) - \nabla_\theta \hat{l}(\theta_T(\phi_2))\|$$

$$+ \|\nabla_\phi^2 \theta_T(\phi_1) - \nabla_\phi^2 \theta_T(\phi_2)\| \|\nabla_\theta \hat{l}(\theta_T(\phi_2))\|$$

$$+ \|[\nabla_\phi \theta_T(\phi_1)]^T\| \|\nabla_\phi \nabla_\theta \hat{l}(\theta_T(\phi_1)) - \nabla_\phi \nabla_\theta \hat{l}(\theta_T(\phi_2))\|$$

$$+ \|[\nabla_\phi \theta_T(\phi_1)]^T - [\nabla_\phi \theta_T(\phi_2)]^T\| \|\nabla_\phi \nabla_\theta \hat{l}(\theta_T(\phi_2))\|$$

$$\overset{(i)}{\leq} LL_{\theta T} \|\theta_T(\phi_1) - \theta_T(\phi_2)\| + M\|\nabla_\phi^2 \theta_T(\phi_1) - \nabla_\phi^2 \theta_T(\phi_2)\|$$

$$+ M_{\theta T}\|\nabla_\phi \nabla_\theta \hat{l}(\theta_T(\phi_1)) - \nabla_\phi \nabla_\theta \hat{l}(\theta_T(\phi_2))\| + L_{\theta T}\|\phi_1 - \phi_2\| M_{\theta T} L$$

$$\leq LM_{\theta T} L_{\theta T}\|\phi_1 - \phi_2\| + M\rho_\theta \|\phi_1 - \phi_2\|$$

$$+ (M_{\theta T}^2 \rho + L_{\theta T}L)M_{\theta T}\|\phi_1 - \phi_2\| + M_{\theta T}LL_{\theta T}\|\phi_1 - \phi_2\|$$

$$= (3LM_{\theta T}L_{\theta T} + M\rho_\theta + M_{\theta T}^3 \rho)\|\phi_1 - \phi_2\| = \rho_{g T}\|\phi_1 - \phi_2\|,$$

where $(i)$ follows from Lemma 2 and 5.

□

**Lemma 8.** *If we assume $\theta_0^1(\phi) = \theta_0^2(\phi)$, based on Assumption 3 and 4, we obtain*

$$\|\theta_T^1(\phi) - \theta_T^2(\phi)\| \leq \sigma_{\theta T},$$

*where $T$ is the iteration number and $\sigma_{\theta T} = (1 + M_{m1}L)^T \frac{\Delta_{12}}{L} - \frac{\Delta_{12}}{L}$.*

*Proof.* Based on the iterative process of $\theta_t(\phi)$, we obtain

$$\|\theta_T^1(\phi) - \theta_T^2(\phi)\|$$

$$\overset{(i)}{\leq} \|\theta_{T-1}^1(\phi) + m(\nabla_\theta \hat{l}_1(\theta_{T-1}^1), \phi) - \theta_{T-1}^2(\phi) - m(\nabla_\theta \hat{l}_2(\theta_{T-1}^2), \phi)\|$$

$$\overset{(ii)}{\leq} \|\theta_{T-1}^1(\phi) - \theta_{T-1}^2(\phi)\| + M_{m1}\|\nabla_\theta \hat{l}_1(\theta_{T-1}^1) - \nabla_\theta \hat{l}_2(\theta_{T-1}^2)\|$$

$$\leq \|\theta_{T-1}^1(\phi) - \theta_{T-1}^2(\phi)\| + M_{m1}\|\nabla_\theta \hat{l}_1(\theta_{T-1}^1) - \nabla_\theta \hat{l}_2(\theta_{T-1}^1)\|$$

$$+ M_{m1}\|\nabla_\theta \hat{l}_2(\theta_{T-1}^1) - \nabla_\theta \hat{l}_2(\theta_{T-1}^2)\|$$

$$\overset{(iii)}{\leq} (1 + M_{m1}L)\|\theta_{T-1}^1(\phi) - \theta_{T-1}^2(\phi)\| + M_{m1}\Delta_{12},$$

where $(i)$ follows from Equation (1), $(ii)$ follows from Assumption 3, $(iii)$ follows from Assumption 4. If we iterate above inequalities from $t = 0$ to $T - 1$, then we obtain:

$$\|\theta_T^1(\phi) - \theta_T^2(\phi)\| \leq (1 + M_{m1}L)^T \frac{\Delta_{12}}{L} - \frac{\Delta_{12}}{L} = \sigma_{\theta T}.$$

□

**Lemma 9.** *Based on Assumptions 3 and 4, Lemma 8, we have following inequality:*

$$\|C_i^1 - C_i^2\| \leq \Delta_{Ci},$$

*where $C_i^j = \nabla_2 m(\nabla_\theta \hat{l}_j(\theta_i), \phi)$ $(i = 0 : T, j \in \{1, 2\})$ and $\Delta_{Ci} = L_{m1}(1 + M_{m1}L)^i \Delta_{12}$.*

*Proof.* Based on the definition of $C_i^j$, we obtain

$$\|C_i^1 - C_i^2\| = \|\nabla_2 m(\nabla_\theta \hat{l}_1(\theta_i^1), \phi) - \nabla_2 m(\nabla_\theta \hat{l}_2(\theta_i^2), \phi)\|$$

$$\leq L_{m1}\|\nabla_\theta \hat{l}_1(\theta_i^1) - \nabla_\theta \hat{l}_2(\theta_i^1) + \nabla_\theta \hat{l}_2(\theta_i^1) - \nabla_\theta \hat{l}_2(\theta_i^2)\|$$

$$\overset{(i)}{\leq} L_{m1}\Delta_{12} + L_{m1}L\|\theta_i^1 - \theta_i^2\|$$

$$\overset{(ii)}{\leq} L_{m1}(1 + M_{m1}L)^i \Delta_{12} = \Delta_{Ci},$$

where $(i)$ follows from Assumption 3 and 4, $(ii)$ follows from Lemma 8.

□

**Lemma 10.** *Then based on Assumptions 3 and 4, Lemma 8, we have following inequality:*

$$\|D_i^1 - D_i^2\| \le \Delta_{Di},$$

*where* $D_i^j = \nabla_j m(\nabla_\theta \hat{l}_j(\theta_i^j), \phi)\nabla_\theta^2 \hat{l}_j(\theta_i^j)$ $(i = 0 : T, j \in 1, 2)$, $\Delta_{Di} = M_{m1}(\rho\sigma_{\theta_i} + \widetilde{\Delta}_{12}) + L_{m1}L(1 + M_{m1}L)^i\Delta_{12}$ *and* $\sigma_{\theta_i}$ *is defined in Lemma 8.*

*Proof.* Based on the definition of $D_i^j$, we obtain

$$\begin{aligned}
\|D_i^1 - D_i^2\| =& \|\nabla_1 m(\nabla_\theta \hat{l}_1(\theta_i^1), \phi)\nabla_\theta^2 \hat{l}_1(\theta_i^1) - \nabla_1 m(\nabla_\theta \hat{l}_2(\theta_i^2), \phi)\nabla_\theta^2 \hat{l}_2(\theta_i^2)\| \\
=& \|\nabla_1 m(\nabla_\theta \hat{l}_1(\theta_i^1), \phi)\nabla_\theta^2 \hat{l}_1(\theta_i^1) - \nabla_1 m(\nabla_\theta \hat{l}_1(\theta_i^1), \phi)\nabla_\theta^2 \hat{l}_2(\theta_i^2) \\
& + \nabla_1 m(\nabla_\theta \hat{l}_1(\theta_i^1), \phi)\nabla_\theta^2 \hat{l}_2(\theta_i^2) - \nabla_1 m(\nabla_\theta \hat{l}_2(\theta_i^2), \phi)\nabla_\theta^2 \hat{l}_2(\theta_i^2)\| \\
\le& \|\nabla_1 m(\nabla_\theta \hat{l}_1(\theta_i^1), \phi)\|\|\nabla_\theta^2 \hat{l}_1(\theta_i^1) - \nabla_\theta^2 \hat{l}_2(\theta_i^2)\| \\
& + \|\nabla_1 m(\nabla_\theta \hat{l}_1(\theta_i^1), \phi) - \nabla_1 m(\nabla_\theta \hat{l}_2(\theta_i^2), \phi)\|\|\nabla_\theta^2 \hat{l}_2(\theta_i^2)\| \\
\le& M_{m1}\|\nabla_\theta^2 \hat{l}_1(\theta_i^1) - \nabla_\theta^2 \hat{l}_2(\theta_i^2) + \nabla_\theta^2 \hat{l}_1(\theta_i^2) - \nabla_\theta^2 \hat{l}_2(\theta_i^2)\| \\
& + L_{m1}L\|\nabla_\theta \hat{l}_1(\theta_i^1) - \nabla_\theta \hat{l}_1(\theta_i^2) + \nabla_\theta \hat{l}_1(\theta_i^2) - \nabla_\theta \hat{l}_2(\theta_i^2)\| \\
\le& M_{m1}(\rho\|\theta_i^1 - \theta_i^2\| + \widetilde{\Delta}_{12}) + L_{m1}L(L\sigma_{\theta i} + \Delta_{12}) \\
\overset{(i)}{\le}& M_{m1}(\rho\sigma_{\theta_i} + \widetilde{\Delta}_{12}) + L_{m1}L(1 + M_{m1}L)^i\Delta_{12} = \Delta_{Di},
\end{aligned}$$

where $(i)$ follow from Lemma 8. $\qquad\square$

**Lemma 11.** *Based on Assumptions 3, 4 and Lemma 1, we obtain*

$$\begin{aligned}
& \|\nabla_\phi \theta_T^1(\phi) - \nabla_\phi \theta_T^2(\phi)\| \\
& \le \sum_{i=0}^{T-1}\left((1 + M_{m1}L)^{T-i-1}\Delta_{Ci} + M_{m2}(1 + M_{m1}L)^{T-i-2}\sum_{j=i+1}^{T-1}\Delta_{Dj}\right),
\end{aligned}$$

*where* $\Delta_{Ci}$ *and* $\Delta_{Dj}$ *have been defined in Lemmas 9 and 10.*

*Proof.* Based on the Lemma 1, we obtain

$$\begin{aligned}
& \|\nabla_\phi \theta_T^1(\phi) - \nabla_\phi \theta_T^2(\phi)\| \\
& \overset{(i)}{\le} \sum_{i=0}^{T-1}\left\|\prod_{j=i+1}^{T-1}(I + D_{T+i-j}^1)C_i^1 - \prod_{j=i+1}^{T-1}(I + D_{T+i-j}^1)C_i^2 + \prod_{j=i+1}^{T-1}(I + D_{T+i-j}^1)C_i^2\right. \\
& \quad\left. - \prod_{j=i+1}^{T-1}(I + D_{T+i-j}^2)C_i^2\right\| \\
& \le \sum_{i=0}^{T-1}\left(\left\|\prod_{j=i+1}^{T-1}(I + D_{T+i-j}^1)\right\|\|C_i^1 - C_i^2\| + \left\|\prod_{j=i+1}^{T-1}(I + D_{T+i-j}^1)\right.\right. \\
& \quad\left.\left. - \prod_{j=i+1}^{T-1}(I + D_{T+i-j}^2)\right\|\|C_i^2\|\right) \\
& \overset{(ii)}{\le} \sum_{i=0}^{T-1}\left((1 + M_{m1}L)^{T-i-1}\|C_i^1 - C_i^2\| + M_{m2}(1 + M_{m1}L)^{T-i-2}\right. \\
& \quad\left.\sum_{j=i+1}^{T-1}\|D_{T+i-j}^1 - D_{T+i-j}^2\|\right) \\
& \overset{(iii)}{\le} \sum_{i=0}^{T-1}\left((1 + M_{m1}L)^{T-i-1}\Delta_{Ci} + M_{m2}(1 + M_{m1}L)^{T-i-2}\sum_{j=i+1}^{T-1}\Delta_{Dj}\right),
\end{aligned}$$

where $(i)$ follows from the definitions that $D_i^j = \nabla_j m(\nabla_\theta \hat{l}_j(\theta_i^j), \phi) \nabla_\theta^2 \hat{l}_j(\theta_i^j)$, $C_i^j = \nabla_2 m(\nabla_\theta \hat{l}_j(\theta_i), \phi)$, $(ii)$ follows from Assumption 3 and $(iii)$ follows from Lemma 9 and 10. □

**Lemma 12.** *(Correspond to Proposition 1) Based on Assumptions 3 and 4, Lemmas 8 and 11, we obtain*

$$\|\nabla_\phi \hat{g}_T^1(\phi) - \nabla_\phi \hat{g}_T^2(\phi)\| = \mathcal{O}(TQ^{T-1}\widetilde{\Delta}_{12} + Q^{2T-1}\Delta_{12}),$$

*where* $Q = 1 + M_{m1}L$.

*Proof.* We first consider $\Delta_{Ci}$ and $\Delta_{Di}$, we obtain

$$\Delta_{Ci} = L_{m1}(1 + M_{m1}L)^i \Delta_{12}) = \mathcal{O}(Q^i \Delta_{12}), \tag{12}$$

$$\Delta_{Di} = \mathcal{O}(M_{m1}(\rho\sigma_{\theta_i} + \widetilde{\Delta}_{12}) + L_{m1}L(1 + M_{m1}L)^i \Delta_{12})$$

$$\overset{(i)}{=} \mathcal{O}(Q^i \Delta_{12} + \widetilde{\Delta}_{12} + Q^i \Delta_{12}) = \mathcal{O}(Q^i \Delta_{12} + \widetilde{\Delta}_{12}), \tag{13}$$

where $(i)$ follows because $\sigma_{\theta_i} = (1 + M_{m1}L)^i \frac{\Delta_{12}}{L} - \frac{\Delta_{12}}{L} = \mathcal{O}(Q^i \Delta_{12})$.

Furthermore, we consider the uniform bound for $\|\nabla_\phi \theta_T^1(\phi) - \nabla_\phi \theta_T^2(\phi)\|$, then we obtain

$$\|\nabla_\phi \theta_T^1(\phi) - \nabla_\phi \theta_T^2(\phi)\|$$

$$\overset{(i)}{=} \mathcal{O}\left(\sum_{i=0}^{T-1}\left(Q^{T-i-2}\left(Q\Delta_{Ci} + \sum_{j=i+1}^{T-1}\Delta_{D(T+i-j)}\right)\right)\right)$$

$$\overset{(ii)}{=} \mathcal{O}\left(\sum_{i=0}^{T-1}\left(Q^{T-i-1}Q^i\Delta_{12} + Q^{T-i-2}\sum_{j=i+1}^{T-1}(Q^{T+i-j}\Delta_{12} + \widetilde{\Delta}_{12})\right)\right)$$

$$= \mathcal{O}\left(\sum_{i=0}^{T-1}\left(Q^{T-1}\Delta_{12} + (T-i-1)Q^{T-i-2}\widetilde{\Delta}_{12} + (Q^{2T-i-2} - Q^{T-1})\Delta_{12}\right)\right)$$

$$= \mathcal{O}\left(\sum_{i=0}^{T-1}((T-i-1)Q^{T-i-2}\widetilde{\Delta}_{12} + Q^{2T-i-2}\Delta_{12})\right)$$

$$\overset{(iii)}{=} \mathcal{O}\left(\sum_{j=0}^{T-1}(jQ^{j-1}\widetilde{\Delta}_{12} + Q^{T+j-1}\Delta_{12})\right)$$

$$= \mathcal{O}\left(TQ^{T-1}\widetilde{\Delta}_{12} + Q^{2T-1}\Delta_{12}\right),$$

where $(i)$ follows from Lemma 11, $(ii)$ follows from Equation (12) and Equation (13), $(iii)$ follows because $j = T - i - 1$. Based on the formulation of $\nabla_\phi \hat{g}_T(\phi)$ in Lemma 6, we have

$$\|\nabla_\phi \hat{g}_T^1(\phi) - \nabla_\phi \hat{g}_T^2(\phi)\| \leq M\|\nabla_\phi \theta_T^1(\phi) - \nabla_\phi \theta_T^2(\phi)\| + M_{\theta T}Q^T \Delta_{12}$$

$$\overset{(i)}{=} \mathcal{O}(TQ^{T-1}\widetilde{\Delta}_{12} + Q^{2T-1}\Delta_{12}),$$

where $(i)$ follows because $M_{\theta T}$ defined in Lemma 2 satisfies that $M_{\theta T} = \mathcal{O}(Q^{T-1})$. □

**Lemma 13.** *Based on the Assumption 3 and Lemma 2, we obtain*

$$\|g_T(\phi_1) - g_T(\phi_2)\| \leq M_{g_T}\|\phi_1 - \phi_2\|,$$

*where* $M_{g_T} = MM_{\theta T}$ *and* $M_{\theta T}$ *is defined in Lemma 2.*

*Proof.* Based on the definition of $g_T(\phi)$, we have

$$\|g_T(\phi_1) - g_T(\phi_2)\| = \|l(\theta_T(\phi_1)) - l(\theta_T(\phi_2))\|$$

$$\overset{(i)}{\leq} M\|\theta_T(\phi_1) - \theta_T(\phi_2)\|$$

$$\overset{(ii)}{\leq} MM_{\theta T}\|\phi_1 - \phi_2\|$$
$$= M_{g_T}\|\phi_1 - \phi_2\|,$$

where $(i)$ is based on Assumption 3, $(ii)$ is based on Lemma 2. $\qquad\square$

**Lemma 14.** *Based on the proposition 1 in* Fallah et al. (2021), *Assumptions 1 and 3, if we set* $\alpha \leq \min\{\frac{1}{2L}, \frac{\mu}{8\rho_{g_T}M_{g_T}}\}$, $\beta_k = \min(\beta, \frac{8}{\mu(k+1)})$ *for* $\beta \leq \frac{8}{\mu}$ *in Algorithm 1, then we have*

$$\mathbb{E}\|\widetilde{\phi}_{MK}^1 - \widetilde{\phi}_{M*}^1\|^2 \leq \mathcal{O}(1)\frac{M_{g_T}^2(1 + \frac{1}{\beta\mu})}{\mu^3}\left(\frac{L_{g_T} + \rho_{g_T}\alpha M_{g_T}}{K} + \frac{M_{g_T}}{\sqrt{K}}\right),$$

*where* $M_{g_T}$ *is defined in Lemma 13,* $L_{g_T}$ *is defined in Lemma 6,* $\rho_{g_T}$ *is defiend in Lemma 7.*

*Proof.* Based on the Proposition 1 in Fallah et al. (2021), we obtain

$$\mathbb{E}\left[\hat{G}_T^1(\widetilde{\phi}_{MK}^1) - \hat{G}_T^1(\widetilde{\phi}_{M*}^1)\right] \leq \mathcal{O}(1)\frac{M_{g_T}^2(1 + \frac{1}{\beta\mu})}{\mu^2}\left(\frac{L_{g_T} + \rho_{g_T}\alpha M_{g_T}}{K} + \frac{M_{g_T}}{\sqrt{K}}\right),$$

where $\hat{G}_T(\phi)$ is defined in Equation (5). Based on the Assumption 1 and the fact that $\widetilde{\phi}_{M*}^1 = \arg\min_\phi \hat{G}_T^1(\phi)$, we have

$$\mathbb{E}\|\widetilde{\phi}_{MK}^1 - \widetilde{\phi}_{M*}^1\|^2 \leq \frac{2}{\mu}\mathbb{E}\left(\hat{G}_T^1(\widetilde{\phi}_{MK}^1) - \hat{G}_T^1(\widetilde{\phi}_{M*}^1)\right)$$
$$\leq \mathcal{O}(1)\frac{M_{g_T}^2(1 + \frac{1}{\beta\mu})}{\mu^3}\left(\frac{L_{g_T} + \rho_{g_T}\alpha M_{g_T}}{K} + \frac{M_{g_T}}{\sqrt{K}}\right).$$

$\qquad\square$

**Lemma 15.** *Based on Assumption 1 and Lemma 13, we have*

$$\|\widetilde{\phi}_*^1 - \phi_*^1\| \leq \frac{2\sqrt{2}MM_{\theta T}}{\mu\sqrt{\delta N}},$$

*where* $N$ *is the sample size.*

*Proof.* Based on Assumption 1 and Lemma 13, from Theorem 2 in Shalev-Shwartz et al. (2010), with probability at least $1 - \delta$, we have

$$g_T^1(\widetilde{\phi}_*^1) - g_T^1(\phi_*^1) \leq \frac{4M_{g_T}^2}{\delta\mu N}.$$

Furthermore, based on Assumption 1 and the fact that $\phi_*^1 = \arg\min g_T^1(\phi)$, we obtain

$$\|\widetilde{\phi}_*^1 - \phi_*^1\|^2 \leq \frac{2}{\mu}\left(g_T^1(\widetilde{\phi}_*^1) - g_T^1(\phi_*^1)\right) \leq \frac{2}{\mu}\frac{4M_{g_T}^2}{\delta\mu N} = \frac{8M_{g_T}^2}{\delta\mu^2 N}.$$

We take the square root from both side and obtain:

$$\|\widetilde{\phi}_*^1 - \phi_*^1\| \leq \frac{2\sqrt{2}M_{g_T}}{\mu\sqrt{\delta N}},$$

with probability at least $1 - \delta$. $\qquad\square$

### A.3 PROOF OF THEOREM 1

Based on our definition of generalization error for the algorithm,

$$g_T^3(\widetilde{\phi}_{MK}^1 - \alpha\nabla_\phi \hat{g}_T^2(\widetilde{\phi}_{MK}^1)) - g_T^3(\phi_*^3)$$
$$\overset{(i)}{\leq} g_T^3(\widetilde{\phi}_{MK}^1 - \widetilde{\phi}_{M*}^1 + \widetilde{\phi}_*^1 + \alpha\nabla_\phi \hat{g}_T^1(\widetilde{\phi}_{M*}^1) - \alpha\nabla_\phi \hat{g}_T^2(\widetilde{\phi}_{MK}^1)) - g_T^3(\phi_*^3)$$

$$\overset{(ii)}{\le} M_{g_T}\|\widetilde{\phi}^1_{MK} - \widetilde{\phi}^1_{M*} + \widetilde{\phi}^1_* - \phi^1_* + \phi^1_* - \phi^3_* + \alpha\nabla_\phi \hat{g}^1_T(\widetilde{\phi}^1_{M*}) - \alpha\nabla_\phi \hat{g}^2_T(\widetilde{\phi}^1_{MK})\| \tag{14}$$

$$\le M_{g_T}\|\widetilde{\phi}^1_{MK} - \widetilde{\phi}^1_{M*}\| + M_{g_T}\|\widetilde{\phi}^1_* - \phi^1_*\| + M_{g_T}\|\phi^1_* - \phi^3_*\|$$
$$+ M_{g_T}\alpha\|\nabla_\phi \hat{g}^1_T(\widetilde{\phi}^1_{M*}) - \nabla_\phi \hat{g}^1_T(\widetilde{\phi}^1_{MK})\| + M_{g_T}\alpha\|\nabla_\phi \hat{g}^1_T(\widetilde{\phi}^1_{MK}) - \nabla_\phi \hat{g}^2_T(\widetilde{\phi}^1_{MK})\|$$
$$\le (M_{g_T} + M_{g_T}L_{g_T}\alpha)\|\widetilde{\phi}^1_{MK} - \widetilde{\phi}^1_{M*}\| + M_{g_T}\|\widetilde{\phi}^1_* - \phi^1_*\| + M_{g_T}\|\phi^1_* - \phi^3_*\|$$
$$+ M_{g_T}\alpha\|\nabla_\phi \hat{g}^1_T(\widetilde{\phi}^1_{MK}) - \nabla_\phi \hat{g}^2_T(\widetilde{\phi}^1_{MK})\|,$$

where $(i)$ follows from Assumption 2, $(ii)$ follows from Lemma 13.

Furthermore, considering Algorithm 1, if we set $\alpha \le \min\{\frac{1}{2L}, \frac{\mu}{8\rho_{g_T}M_{g_T}}\}$, $\beta_k = \min(\beta, \frac{8}{\mu(k+1)})$ for $\beta \le \frac{8}{\mu}$, based on Lemma 12, 14, 15, with probability at least $1 - \delta$, we obtain

$$\mathbb{E}[g^3_T(\widetilde{\phi}^1_{MK} - \alpha\nabla_\phi \hat{g}^2_T(\widetilde{\phi}^1_{MK})) - g^3_T(\phi^3_*)]$$

$$\le (M_{g_T} + M_{g_T}L_{g_T}\alpha)\left\|\mathcal{O}(1)\frac{M_{g_T}}{\mu^2}\sqrt{\frac{L_{g_T} + \rho_{g_T}\alpha M_{g_T}}{\beta K} + \frac{M_{g_T}}{\beta\sqrt{K}}}\right\| + M_{g_T}\left\|\frac{2\sqrt{2}M_{g_T}}{\mu\sqrt{\delta N}}\right\|$$
$$+ M_{g_T}\delta_{13} + M_{g_T}\alpha\mathcal{O}(TQ^{T-1}\widetilde{\Delta}_{12} + Q^{2T-1}\Delta_{12}),$$

where $\delta_{13} = \|\phi^1_* - \phi^3_*\|$, $Q = (1 + M_{m1}L)$, $K$ is the step number for update, $N$ is the sample size for training.

Then for Lipschitz term $M_{g_T}$ defined in Lemma 13,

$$M_{g_T} = MM_{\theta T} = \mathcal{O}(Q^{T-1}),$$

where $M_{\theta T}$ defined in Lemma 2 satisfies $M_{\theta T} = \mathcal{O}(Q^{T-1})$.

For Lipschitz term $L_{g_T}$ defined in Lemma 6, we first compute the order for $L_{\theta T}$ which is defined in Lemma 5, then we obtain

$$L_{\theta T} = \mathcal{O}\left(\sum_{i=0}^{T-1} Q^{T-i-1}M_{A_i} + \sum_{i=0}^{T-1} Q^{T-i-2}\sum_{j=i+1}^{T-1} M_{B(T+i-j)}\right)$$

$$= \mathcal{O}\left(\sum_{i=0}^{T-1} Q^{T-i-1}Q^{i-1} + \sum_{i=0}^{T-1} Q^{T-i-2}\sum_{j=i+1}^{T-1} Q^{T+i-j-1}\right)$$

$$\overset{(i)}{=}\mathcal{O}\left(\sum_{i=0}^{T-1} Q^{T-2} + \sum_{i=0}^{T-1} Q^{T-i-2}Q^{T-1}\right) = \mathcal{O}(TQ^{T-2} + Q^{2T-2}),$$

where $(i)$ follows from Lemmas 3 and 4. Then, we obtain

$$L_{g_T} = ML_{\theta T} + LM^2_{\theta T}$$
$$= \mathcal{O}(TQ^{T-2} + Q^{2T-2} + Q^{2T-2}) = \mathcal{O}(TQ^{T-2} + Q^{2T-2}).$$

For Lipschitz term $\rho_{g_T}$ defined in Lemma 7, we have

$$\rho_{g_T} = 3LM_{\theta T}L_{\theta T} + M\rho_\theta + M^3_{\theta T}\rho = \mathcal{O}(TQ^{2T-3} + Q^{2T-3}).$$

Then, the proof is complete.

## A.4   PROOF OF REMARK 2

In terms of M-L2O generalization error, based on the Equation (14) in Appendix A.3, we have

$$g^3_T(\widetilde{\phi}^1_{MK} - \alpha\nabla_\phi \hat{g}^2_T(\widetilde{\phi}^1_{MK})) - g^3_T(\phi^3_*)$$
$$\le M_{g_T}\|\widetilde{\phi}^1_{MK} - \widetilde{\phi}^1_{M*} + \widetilde{\phi}^1_* - \phi^1_* + \phi^1_* - \phi^3_* + \alpha\nabla_\phi \hat{g}^1_T(\widetilde{\phi}^1_{M*}) - \alpha\nabla_\phi \hat{g}^2_T(\widetilde{\phi}^1_{MK})\|$$
$$\le M_{g_T}\|\widetilde{\phi}^1_{MK} - \widetilde{\phi}^1_{M*}\| + M_{g_T}\|\widetilde{\phi}^1_* - \phi^1_*\| + M_{g_T}\alpha\|\nabla_\phi \widetilde{g}^2_T(\widetilde{\phi}^1_{MK}) - \nabla_\phi \widetilde{g}^1_T(\widetilde{\phi}^1_{M*})\| + M_{g_T}\delta_{13},$$

where $\delta_{13} = \|\phi_*^1 - \phi_*^3\|$.

In terms of Transfer Learning L2O generalization error with learned initial point $\widetilde{\phi}_K$, we have

$$
\begin{aligned}
& g_T^3(\widetilde{\phi}_K^1 - \alpha\nabla_\phi\hat{g}_T^2(\widetilde{\phi}_K^1)) - g_T^3(\phi_*^3) \\
& \leq M_{g_T}\|\widetilde{\phi}_K^1 - \alpha\nabla_\phi\hat{g}_T^2(\widetilde{\phi}_K^1) - \phi_*^3\| \\
& \overset{(i)}{=} M_{g_T}\|\widetilde{\phi}_K^1 - \alpha\nabla_\phi\hat{g}_T^2(\widetilde{\phi}_K^1) - (\widetilde{\phi}_{M*}^1 - \alpha\nabla_\phi\hat{g}_T^1(\widetilde{\phi}_{M*}^1)) + \widetilde{\phi}_*^1 - \phi_*^3\| \\
& \leq M_{g_T}\|\widetilde{\phi}_K^1 - \widetilde{\phi}_{M*}^1\| + M_{g_T}\|\widetilde{\phi}_*^1 - \phi_*^1\| + M_{g_T}\alpha\|\nabla_\phi\widetilde{g}_T^2(\widetilde{\phi}_K^1) - \nabla_\phi\widetilde{g}_T^1(\widetilde{\phi}_{M*}^1)\| + M_{g_T}\delta_{13},
\end{aligned}
$$

where $(i)$ follows from $\widetilde{\phi}_*^1 = \widetilde{\phi}_{M*}^1 - \alpha\nabla_\phi\hat{g}_T^1(\widetilde{\phi}_{M*}^1)$, $\delta_{13} = \|\phi_*^1 - \phi_*^3\|$. Then, the proof is complete.

## A.5 PROOF OF EQ. 8 IN SUBSECTION 5.3

We assume that $\widetilde{\phi}_*^3 = \widetilde{\phi}_{M*}^3 - \alpha\nabla_\phi\hat{g}_T^3(\widetilde{\phi}_{M*}^3)$, then we have

$$
\begin{aligned}
& g_T^3(\widetilde{\phi}_{MK}^1 - \alpha\nabla_\phi\hat{g}_T^2(\widetilde{\phi}_{MK}^1)) - g_T^3(\phi_*^3) \\
& \leq M_{g_T}\|\widetilde{\phi}_{MK}^1 - \alpha\nabla_\phi\hat{g}_T^2(\widetilde{\phi}_{MK}^1) - \phi_*^3\| \\
& \leq M_{g_T}\|\widetilde{\phi}_{MK}^1 - \alpha\nabla_\phi\hat{g}_T^2(\widetilde{\phi}_{MK}^1) - \widetilde{\phi}_{M*}^3 + \alpha\nabla_\phi\hat{g}_T^3(\widetilde{\phi}_{M*}^3) + \widetilde{\phi}_*^3 - \phi_*^3\| \\
& \leq M_{g_T}\|\widetilde{\phi}_{MK}^1 - \widetilde{\phi}_{M*}^1\| + M_{g_T}\|\widetilde{\phi}_*^3 - \phi_*^3\| + M_{g_T}\alpha\|\nabla_\phi\hat{g}_T^2(\widetilde{\phi}_{MK}^1) - \nabla_\phi\hat{g}_T^3(\widetilde{\phi}_{M*}^3)\| \\
& \quad + M_{g_T}\|\widetilde{\phi}_{M*}^1 - \widetilde{\phi}_{M*}^3\|.
\end{aligned}
$$

Then, the proof is complete.

## A.6 ADDITIONAL EXPERIMENTS

**New Optimizees: Rosenbrock** We conduct additional experiments with substantially different optimizees, *i.e.* Rosenbrock (Rosenbrock, 1960). In this case, the optimizes are required to minimize a two-dimensional non-convex function taking the following formulation:

$$
f(x, y) = (x - 1)^2 + 100(y - x^2)^2, \tag{15}
$$

which is challenging for algorithms to converge to the global minimum (Tani et al., 2021).

We specify $D_{\text{adapt}}$ and $D_{\text{test}}$ to be the family of Rosenbrock optimizees with randomly sampled initial points from standard normal distribution. In contrast, the training optimizees are still LASSO with a mixture of uniform distribution from which the coefficient matrices are sampled. The experiments are repeated for 10 times, with all the algorithms receiving identical adaptation and testing samples in each run. Figure A5a shows the curves of the logarithm of the objective values generated by different methods, where our proposed M-L2O outperforms other baselines significantly. At 500-th step, the (mean, standard deviation) of the logarithmic objective values for {Vanilla L2O, TL, DT, M-L2O} are $\{(0.977, 0.225), (-2.170, 1.312), (-4.864, 0.395), (-6.832, 0.445)\}$, which provides numerical supports of the advantage of our methods.

**New Evaluation: Interpolation**

To obtain new optimize weights, we employ a linear interpolation strategy between two adapted optimizers. The first one is optimized on the optimizees that are similar to those used in training, and the second is optimized on the optimizees that are similar to those used in testing. We introduce a factor $\alpha$ to control the interpolation between the two weights, denoted by $\boldsymbol{w}_1$ and $\boldsymbol{w}_2$, respectively, and caluclate the new weights as follows:

$$
\boldsymbol{w} = \alpha\boldsymbol{w}_1 + (1 - \alpha)\boldsymbol{w}_2.
$$

In Figure A5b, we present the mean values of the logarithmic loss, as well as the 95% confidence interval. The results of TL and M-L2O validate our claim that adapting to training-like optimizes tend to yield better performance than adapting to optimizees that more resemble the testing optimizees.

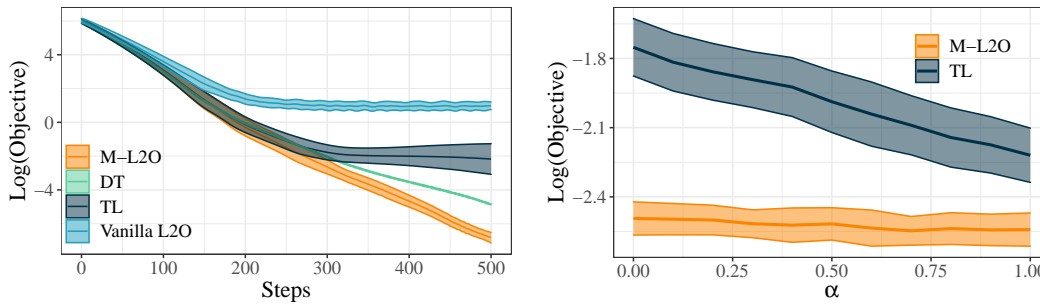

(a) Convergence speeds on Rosenbrock optimizees. We repeat the experiments for 10 times, and present the 95% confidence intervals are shown in the figure.

(b) Convergence speeds on LASSO optimizees, with different interpolation weights $\alpha$. Both the mean and the 95% confidence intervals are shown in the figure.

Figure A5: Visualization of additional experiment results.

