# OpenReview forum: "M-L2O: Towards Generalizable Learning-to-Optimize by Test-Time Fast Self-Adaptation"
_ICLR.cc/2023/Conference — ICLR 2023 poster_

### Official Review · Reviewer_2sEx · 2022-10-16

**Confidence:** 3
**Correctness:** 4
**Technical Novelty And Significance:** 2
**Empirical Novelty And Significance:** 1
**Recommendation:** 5

**Clarity, Quality, Novelty And Reproducibility:**

While the paper is mathematically intensive, the assumptions and results are stated clearly. I would encourage the authors to comment on when the assumptions in the theoretical results can be assumed to hold.

While the paper is well written, I think the motivation could be improved. Where would the proposed method be the most useful? Currently, the experiments are small toy examples that do not demonstrate any practical utility of the method. Could more experiments with neural networks be added? If not, maybe more examples where current methods fail could be added.

The novelty of the paper is about average, I don’t know of any previous papers which combine meta-learning and learning to optimize, but combining these two well-known fields is not very novel.

From a reproducibility perspective, adding error bars to the experiment would be helpful.



**Strength And Weaknesses:**

Strengths:

* The paper is relatively well written

Weaknesses:

* The novelty is limited.
* The experiments are very limited, using toy problems and small synthetic datasets.
* The assumptions of convexity might not hold in practice.
* The utility or broader interest of the method is unclear.




**Summary Of The Paper:**

The paper proposes to add meta-adaptation into the learn-to-optimize pipeline. The authors present an algorithm for adapting L2O inspired by MAML in algorithm 1. Under standard assumptions in convex optimization (strong convexity and Lipschitz continuity) the authors then derive generalization bounds of the proposed method. The authors verify the efficacy of the proposed method on LASSO and quadratic objectives functions with small synthetic datasets.


**Summary Of The Review:**

The paper combines meta-learning and learning to optimize, proposing a natural algorithm and deriving some generalization bounds for it. Experimentally, the authors show some benefits on small synthetic tasks with the lasso or quadratic objective. While the paper is technically sound, the paper does little to demonstrate any practical utility or new theoretical insight. Thus, I recommend rejection.

---

> ### Author Response · Authors · 2022-11-15
> **Response Part II**
>
> Many thanks for your expert review!
>
> Q3: The assumptions of convexity might not hold in practice. While the paper is mathematically intensive, the assumptions and results are stated clearly. I would encourage the authors to comment on when the assumptions in the theoretical results can be assumed to hold.
>
> A: Thanks for your suggestion! For the strongly-convex assumption, many theoretical papers [r1,r2] have demonstrated that loss landscape exhibits strongly-convex property under the over-parameterization condition. Furthermore, the latest MAML generalization analysis [r3] also requires such an assumption while our meta learning theory cannot skip it. Other reviewers also agreed that Assumptions 1, 3 and 4 are rather standard in theory papers.
>
> For the Assumption 2, we previously assumed that meta optimal point $\phi_{M\ast}^1$ and original optimal point $\phi_\ast^1$ satisfies following equation $\phi_\ast^1 = \phi_{M\ast}^1-\alpha\nabla_\phi \hat{g}^1_T(\phi_{M\ast}^1)$. However, there always exists a trivial solution ($\phi_{M\ast}^1=\phi_{\ast}^1, \nabla_\phi\hat{g}^1_T(\phi_{\ast}^1)=0$) that satisfies. Hence, we rewrite Assumption 2 and assume that there exists non-trivial ($\phi_{M\ast}^1\neq\phi_{\ast}^1$) meta optimal points $\phi_{M\ast}^1$.  Note that for non-trivial optimal points, the above equation automatically satisfies due to the definition of $\phi_{M\ast}^1$ and the existence of the trivial optimal solution.
>
> In MAML experiments, considering there exists a single training task, then we should expect to learn a well-adapted point (which can adapt to the training task with few adaptations) rather than the task optimal point. Otherwise, MAML will be exactly the same as transfer learning and such a learned point is difficult to transfer when encountering new tasks. Hence, it is natural to make the assumption that there exists non-trivial meta optimal points which we expect to learn and they are not task optimal points (the trivial solution). We have added such comments in the revision.
>
> $\textbf{Note}$: All $\phi$ in above answer refer to $\widetilde{\phi}$ since Openreview shows improperly with the equation containing multiple “widetilde” symbols.
>
> [r1] A Convergence Theory for Deep Learning via Over-Parameterization
>
> [r2] Gradient Descent Provably Optimizes Over-parameterized Neural Networks
>
> [r3] Generalization of model-agnostic meta-learning algorithms: Recurring and unseen tasks
>
> Q4: The utility or broader interest of the method is unclear. While the paper is well written, I think the motivation could be improved. Where would the proposed method be the most useful?
>
> A: Our method has broader application impacts. For one example, M-L2O is suitable for repeatedly solving an optimization problem defined on consistently changed data, which abstracts a lot of industrial applications. Here are a few examples:
> - For a large-scale recommendation system that is usually latency sensitive and has frequently updated user data, it boils down to updating a regression model (often as simple as linear) daily, which could be resolved by fast adaptation with our M-L2O.
> - As indicated in [r1], for personalization purpose on resource-constrained IoT devices, the fast adaptation capability via M-L2O is also desirable
> - Another application involving frequently updated environment variables is the multi-agent financial market simulation. It could also be formulated as L2O with shifting inputs, like studied in [r3].
>
> We thank the reviewer for bringing up the question and will discuss those potential broader impacts in our final paper.
>
> [r1] Learning to Optimize in Swarms
>
> [r2] HALO: Hardware-Aware Learning to Optimize
>
> [r3] Calibration of Shared Equilibria in General Sum Partially Observable Markov Games

---

> ### Author Response · Authors · 2022-11-15
> **Response Part I**
>
> Many thanks for your expert review!
>
> Q1: The novelty is limited. The novelty of the paper is about average, I don’t know of any previous papers which combine meta-learning and learning to optimize, but combining these two well-known fields is not very novel.
>
> A: We respectfully argue that “combimng two well-known fields” is an unfair and imprecise assessment of our work, that significantly overlooks our true merit. Our strong novelty lies in two aspects: i) the novel problem setting and ii) insightful results from both empirical and theoretical perspectives.
>
> - *[Novel and Challenging Problem Setting]* We target a painpoint in L2O first time, i.e., **out-of-domain generalization of learned optimizer**, for the general situation. Previous works [r1,r2,r3] show that it is already highly difficult  for L2O to generalize to new test problems even within the same “task distribution” but just with more optimization steps. Our work, as a pioneering study, investigates the out-of-domain transferability of L2O on unseen test cases with a more substantial deviation from the training task distribution.
>
> - *[Empirical and Theoretical Insights]* From the empirical aspects, we politely point out that combining MAML and L2O is NOT straightforward. Both MAML and L2O are formulated with inner and outer loops. We carefully customize MAML into the outer loop of L2O (the optimizer update), bringing superior generalization of learned optimizer. Also we demonstrate training-like adaptation can imply better generalization. Theoretically, we for the first time capture the generic class of L2O without algorithm and framework restrictions, and develop its generalization result, by explicitly showing how loss landscape and algorithm design shape the generalization performance.
>
> [r1] Training Stronger Baselines for Learning to Optimize
>
> [r2] Learned Optimizers that Scale and Generalize
>
> [r3] Learning Gradient Descent: Better Generalization and Longer Horizons
>
> Q2: The experiments are very limited, using toy problems and small synthetic datasets. Currently, the experiments are small toy examples that do not demonstrate any practical utility of the method. Could more experiments with neural networks be added? If not, maybe more examples where current methods fail could be added. From a reproducibility perspective, adding error bars to the experiment would be helpful.
>
>
> A: We have updated non-convex experiments Rosenbrock [r1] in our revised version Appendix A.5. Both convex and non-convex experimental results demonstrate the superiority of M-L2O. We also incorporate the error bars in updated figures as suggested. It can be observed that M-L2O enjoys a smaller error bar compared with other benchmarks.
> The results are shown in the table below.
> | Method | Mean Log Objective | Standard Deviation |
> | :------: | :------: | :------: |
> | Vanilla L2O | 0.977 | 0.225 |
> | DT | -4.864 | 0.395 |
> | TL | -2.170 | 1.312 |
> | M-L2O | -6.832 | 0.445 |
>
> Training L2O on neural networks takes a longer time but we have launched the experiments. We will add those results to the final version of the paper when ready.
>
> [r1]  An automatic method for finding the greatest or least value of a function

---

> ### Author Response · Authors · 2022-11-23
> **Could you please check our response?**
>
> Dear Reviewer 2sEx:
>
> Since the author-reviewer discussion period has started for a few days, we will appreciate it if you could check our response to your review comments. During the discussion stage 1, we have updated non-convex experiments Rosenbrock in our revised version Appendix A.5. The new experimental results further demonstrate the superiority of proposed M-L2O. Meanwhile, the novelty of this paper has been well-recognized by other reviewers (Reviewer 5uH7, Reviewer egHk, Reviewer GEXL). If our response resolves your concerns, we kindly ask you to consider raising the rating of our work. Certainly, we are more than happy to address any further comments that you may have. Thank you very much for your time and efforts!

---

### Official Review · Reviewer_Sfir · 2022-10-23

**Confidence:** 3
**Correctness:** 2
**Technical Novelty And Significance:** 3
**Empirical Novelty And Significance:** 3
**Recommendation:** 5

**Clarity, Quality, Novelty And Reproducibility:**

# Clarity
Overall, the paper is clear to understand. However, in section 3 the notation is sometimes unclear.

# Quality
The authors provide both theoretical and empirical analysis of their algorithm, which is good. The theory appears to be sound (I only skimmed it), although assumption 2 seems a bit strong. The algorithm outperforms several baselines from previous work, although standard errors were not provided and so statistical significance cannot be confirmed.

# Novelty
The authors differentiate between the adaptation and testing optimizees, which I think is an important novel contribution. The algorithm itself is built on MAML, with an additional single gradient step in the objective.

# Reproducibility
Code is not provided. However, the algorithm and benchmarks would be fairly simple to implement, and most of the hyperparameters are provided. Therefore, I don't think reproducibility is an issue.


**Strength And Weaknesses:**

# Strengths
- The presentation of the mathematical results is generally pretty good. The authors clearly state the assumptions and explain the results intuitively.
- This paper makes distinctions between the optimizees used for adaptation and testing, which is unusual in the literature as far as I know. However, I think this is a better approximation of practical situations, where during testing we may have access to small amounts of data that may or may not be drawn from the current distribution of interest.
- The proposed algorithm is a straightforward simple modification of directly applying MAML to L2O, basically adding a gradient step to the objective. The authors provide both theoretical and empirical analysis, and it seems M-L2O empirically outperforms the baselines.

# Weaknesses
- The notation is messy. In section 3.1, it is not clear where the different tasks (line 3 of Alg. 1, $g^1, g^2, g^3$) come in. I think it would be more helpful to explain the general structure of L2O algorithms first. $Q^i$ is not defined in Proposition 1.
- Assumption 2 seems fairly strong. Intuitively it makes sense but I think it would be good to have some evidence, theoretical or empirical.
- As far as I understand it, vanilla L2O does not contain any meta-learning, the optimizer is random. In this case, vanilla L2O would be a misnomer, and the main baselines of interest would be Transfer Learning and Direct Transfer. The figures do suggest that M-L2O may be able to improve over those two algorithms. Therefore, should Remark 2 compare between M-L2O and Transfer Learning?
- Standard errors are not provided in the figures or tables. To judge which algorithm is statistically significantly better, uncertainty estimates would be helpful.

## Minor
- In Theorem 1 and Remark 2, are norm symbols missing from the left hand side of the inequality?
- What form does the optimizer $m$ take in the experiments?


**Summary Of The Paper:**

This paper considers the problem of OOD generalization in L2O, i.e. meta-learning an optimizer that is effective on objectives not seen during training. It proposes an algorithm based on MAML (Finn et al. 2017) and analyzes its generalization ability, measured by the loss at testing time. Unsurprisingly, generalization depends on the difference between the training and testing optimizees, data size, and training length. The authors show that their algorithm M-L2O generalizes better than no meta-learning at all. Experiments are done on quadratic and LASSO benchmarks and confirm the theoretical results. M-L2O also improves on directly applying MAML to L2O.

**Summary Of The Review:**

I think that the proposed algorithm is an interesting approach to tackle OOD generalization for L2O; it is clearly presented and the authors provide both theoretical and empirical analysis. However, there are some points described above (strength of assumption 2, remark 2, standard errors) that raise some questions about the claims made in this paper.

# Update

The majority of the weaknesses I raised were adequately addressed by the authors, so I raised my score from 3 to 5.

---

> ### Author Response · Authors · 2022-11-15
> **Response Part II**
>
> Many thanks for your review!
>
> Q5: In Theorem 1 and Remark 2, are norm symbols missing from the left hand side of the inequality?
>
> A: No norm symbols are missing for Theorem 1 and Remark 2. Generalization error is defined by the difference between optimal optimizer loss and our updated optimizer loss. Loss function outputs are scalars thus no norm symbols needed.
>
> Q6: What form does the optimizer $m$ take in the experiments?
>
> A: In our experiments, we follow [r1] to use LSTM networks as the structure of the optimizer. The LSTM has one layer and a hidden dimension of 32. We have also included more details in Section 5.
>
> [r1] Training Stronger Baselines for Learning to Optimize

---

> > ### Comment · Reviewer_Sfir · 2022-11-18
> > **Updates after rebuttal**
> >
> > After reading the rebuttal and revised paper, I am more positive on this paper. The authors were able to clarify the theory and strengthen the  experimental results. However, the notation is still somewhat messy, and the theoretical results are a bit difficult for readers to parse. Therefore, I am raising my score to 5.

---

> > > ### Author Response · Authors · 2022-11-19
> > > **Thanks for your updates and rasing the score**
> > >
> > > Thanks for your recognition of our further revision and raising the score. In terms of messy notation, if the reviewer can point out specifically, we are more than happy to clarify. Besides, the current theoretical result is the nature of the L2O generalization problem. We have explained the meaning of each generalization term after the theorem. Hopefully, this explanation helps the readers to understand theoretical results easily.

---

> > > ### Author Response · Authors · 2022-11-23
> > > **Further revision update**
> > >
> > > Dear Reviewer Sfir:
> > >
> > > During the discussion period, we have largely revised the wording and theorem explanation. The clarity and readability of this paper has been well recognized by other reviewers (Reviewer egHk, Reviewer 2sEx, Reviewer GEXL). We even include a figure of L2O pipeline in Section 3 to further explain the notation meanings and L2O problem formulation in the revision. If our updated revision resolves your concerns, we kindly ask you to consider raising the rating of our work. Certainly, we are more than happy to address any further comments and specific suggestions that you may have. Thank you very much for your time and efforts!

---

> ### Author Response · Authors · 2022-11-15
> **Response Part I**
>
> Many thanks for your review!
>
> Q1: The notation is messy. In section 3.1, it is not clear where the different tasks (line 3 of Alg. 1, $g^1, g^2, g^3$) come in. I think it would be more helpful to explain the general structure of L2O algorithms first. $Q^i$ is not defined in Proposition 1.
>
> A: Thanks reviewer for pointing this out. We have added the L2O algorithm general structure introduction in the revision as suggested. In short, instead of applying traditional optimizer, e.g., SGD and ADAM to compute parameter updates in general loss function (we call it as $\textbf{optimizee } or \textbf{ task}$), L2O aims to estimate such updates by a model, which we call $\textbf{optimizer}$. In practice, we usually choose LSTM or MLP as an optimizer model. Specifically, optimizer model takes the optimizee information (e.g. loss, gradient) as input and outputs the update estimation. For our algorithm M-L2O, it aims to learn a well-adapted optimizer initial point in training task $g^1$. Then, we make further optimizer initial adaptation with the adaptation task $g^2$. Finally, we apply such an adapted optimizer to train new testing task $g^3$ and evaluate its performance. In Proposition 1, $Q=1+M_{m1}L$ and $i$ denotes the power index. Thus, $Q^i=(1+M_{m1}L)^i$.
>
> Q2: Assumption 2 seems fairly strong. Intuitively it makes sense but I think it would be good to have some evidence, theoretical or empirical.
>
> A: The Assumption 2 is not strong and it is even natural to make from the experimental view. For the Assumption 2, we previously assumed that meta optimal point $\phi_{M\ast}^1$ and original optimal point $\phi_\ast^1$ satisfies following equation $\phi_\ast^1 = \phi_{M\ast}^1-\alpha\nabla_\phi \hat{g}^1_T(\phi_{M\ast}^1)$. However, there always exists a trivial solution ($\phi_{M\ast}^1=\phi_{\ast}^1, \nabla_\phi\hat{g}^1_T(\phi_{\ast}^1)=0$) that satisfies. Hence, we rewrite Assumption 2 and assume that there exists non-trivial ($\phi_{M\ast}^1\neq\phi_{\ast}^1$) meta optimal points $\phi_{M\ast}^1$.  Note that for non-trivial optimal points, the above equation automatically satisfies due to the definition of $\phi_{M\ast}^1$ and the existence of the trivial optimal solution.
>
> In MAML experiments, considering there exists a single training task, then we should expect to learn a well-adapted point (which can adapt to the training task with few adaptations) rather than the task optimal point. Otherwise, MAML will be exactly the same as transfer learning and such a learned point is difficult to transfer when encountering new tasks. Hence, it is natural to make the assumption that there exists non-trivial meta optimal points which we expect to learn and they are not task optimal points (the trivial solution).
>
> Q3: As far as I understand it, vanilla L2O does not contain any meta-learning, the optimizer is random. In this case, vanilla L2O would be a misnomer, and the main baselines of interest would be Transfer Learning and Direct Transfer. The figures do suggest that M-L2O may be able to improve over those two algorithms. Therefore, should Remark 2 compare between M-L2O and Transfer Learning?
>
> A: Thanks for the reviewer’s suggestion! To compare the M-L2O and Transfer Learning in Remark2, we just need to change the vanilla L2O initial point $\phi$ with Transfer Learning initial point $\phi_K^1$. Then, the generalization comparison will be between $||\phi_K^1-\phi_{M\ast}^1||$  (TL) and $||\phi_{MK}^1-\phi_{M\ast}^1||$ (M-L2O). Note that $\phi_{MK}^1$ is trained to converge to $\phi_{M\ast}^1$ as $K$ increases thus M-L2O term converges while TL not. Such a theoretical result demonstrates the superiority of M-L2O. The reason is that M-L2O outputs a meta-trained initial point and benefits from adaptation procedure. We have rewritten Remark 2 as suggested in the revision.
>
>
> $\textbf{Note}$: All $\phi$ in above answer2 and answer3 refer to $\widetilde{\phi}$ since Openreview shows improperly with the equation containing multiple “widetilde” symbols.
>
> Q4: Standard errors are not provided in the figures or tables. To judge which algorithm is statistically significantly better, uncertainty estimates would be helpful.
>
> A: Thanks for this suggestion. We have updated all figures with standard errors in the revision. The detailed results are listed in Table 1 in Section 5. Both table and figures show that M-L2O enjoys a smaller standard error as well as loss compared with other benchmarks, which further demonstrate the superiority of the proposed algorithm.

---

### Official Review · Reviewer_5uH7 · 2022-10-23

**Confidence:** 4
**Correctness:** 2
**Technical Novelty And Significance:** 2
**Empirical Novelty And Significance:** Not applicable
**Recommendation:** 8

**Clarity, Quality, Novelty And Reproducibility:**

The clarity of this work needs large improvement, as the reviewer finds the major contributions and introduction of L2O really hard to follow. Because of the lack of clarity, the reviewer cannot determine the novelty of this work. The proof seems to be reasonable (although the reviewer cannot follow the proof exactly due to the clarity issue), but it seems to be a standard convergence analysis of convex optimizations with different assumptions (Lipschitz continuity, strongly convexity, and some stochasticity). The numerical experiments seem to be reproducible.

**Strength And Weaknesses:**

Strength:
1. The theoretical analysis looks solid
2. The numerical experiments corroborate the theoretical results.

Weakness:
1. The L2O part problem definition needs more descriptions – in section 3.1, it would be much better to introduce the different notations with better characterizations:  (a) It would be better to provide more description on the relationship between $\phi$ and $\theta_t$.
(b) Is $\xi_j$ a vector data sample or it is a parameter that indexed the task number?
(c) What is $z_t$?
(d) In the sentence above equation, $\zeta_t$ is referred to as a data sample, while in the first sentence of section 3.1, $\xi_t$ is referred to as a data sample, what is the relationship between $\zeta_t$ and $\xi_t$?
(e) Where does equation 2 come from? Perhaps the author can provide more introduction/motivation on why $\nabla_\phi \theta_T(\phi)$ has the form in equation (2)?
2. The intuition of M-L2O is not clearly stated – why is the empirical risk (equation 5) related to MAML?
3. The notations in section 3.2 are confusing – where do $g_T^1,g_T^2,g_T^3$ appear in Algorithm 1?
4. The assumptions need clarifications:
(a) (Minor) Assumptions 1, 3, and 4 seem to be standard, but it would be better to address their appearance in some prior papers.
(b) (Major) Why can we directly assume $\widetilde{\phi}^1_*$ and $\widetilde{\phi}^1_{M*}$ satisfy the conditions in assumption 2?
5. The main result (theorem 1) is a bit confusing – the last error term in Theorem 1 has an exponential dependency on $Q$, and the author proposes to set $\alpha\leq \mathcal{O}(1/TQ^{3T-4}+Q^{4T-4})$ to avoid such exponential dependency, which implies that $\alpha$ should be exponentially small. And if the reviewer understands correctly, the new $\alpha$ term seems to be the major novelty of this paper (as suggested in equation 5), so does this mean that the proposed M-L2O actually does not differ too much from the original L2O since we will be setting $\alpha\to0$ anyway?


**Summary Of The Paper:**

This paper proposes the M-L2O algorithm, as a substitute for the L2O algorithm, and then provides theoretical analysis and numerical experiments of the M-L2O algorithm.

**Summary Of The Review:**

In summary, the reviewer cannot provide a clear evaluation of the quality of this paper due to the lack of introduction on the problem formulation, assumptions,  and clarity on the notations. The reviewer encourages the authors to largely rewrite the paper to improve the clarity on the aforementioned issue in the rebuttal.

---

> ### Author Response · Authors · 2022-11-15
> **Response Part II**
>
> Many thanks for your expert review!
>
> Q4: The assumptions need clarifications: (a) (Minor) Assumptions 1, 3, and 4 seem to be standard, but it would be better to address their appearance in some prior papers. (b) (Major) Why can we directly assume and satisfy the conditions in assumption 2?
>
> A: (a) We have discussed the mentioned assumptions in prior papers in the revision.
>
> (b) For the Assumption 2, we previously assumed that meta optimal point $\phi_{M\ast}^1$ and original optimal point $\phi_\ast^1$ satisfies following equation $\phi_\ast^1 = \phi_{M\ast}^1-\alpha\nabla_\phi \hat{g}^1_T(\phi_{M\ast}^1)$. However, there always exists a trivial solution ($\phi_{M\ast}^1=\phi_{\ast}^1, \nabla_\phi\hat{g}^1_T(\phi_{\ast}^1)=0$) that satisfies. Hence, we rewrite Assumption 2 and assume that there exists non-trivial ($\phi_{M\ast}^1\neq\phi_{\ast}^1$) meta optimal points $\phi_{M\ast}^1$.  Note that for non-trivial optimal points, the above equation automatically satisfies due to the definition of $\phi_{M\ast}^1$ and the existence of the trivial optimal solution.
>
> In MAML experiments, considering there exists a single training task, then we should expect to learn a well-adapted point (which can adapt to the training task with few adaptations) rather than the task optimal point. Otherwise, MAML will be exactly the same as transfer learning and such a learned point is difficult to transfer when encountering new tasks. Hence, it is natural to make the assumption that there exists non-trivial meta optimal points which we expect to learn and they are not task optimal points (the trivial solution).
>
> $\textbf{Note}$: All $\phi$ in above answer refer to $\widetilde{\phi}$ since Openreview shows improperly with the equation containing multiple “widetilde” symbols.
>
> Q5: The main result (theorem 1) is a bit confusing – the last error term in Theorem 1 has an exponential dependency on $Q$, and the author proposes to set $\alpha \leq \mathcal{O}(1/TQ^{3T-4}+Q^{4T-4})$ to avoid such exponential dependency, which implies that  $\alpha$ should be exponentially small. And if the reviewer understands correctly, the new $\alpha$ term seems to be the major novelty of this paper (as suggested in equation 5), so does this mean that the proposed M-L2O actually does not differ too much from the original L2O since we will be setting $\alpha \rightarrow 0$ anyway?
>
> A: $\alpha$ is not close to zero and M-L2O is different from original L2O. Note that $T$ is the optimizee update iteration number per epoch. It has been fixed in L2O experiments (e.g. 5, 10) and different from the epoch number (which is denoted by $K$). Hence, $\alpha$ is a determined constant in the whole training process and will not go to zero as epoch increases.
>
> Q6: The clarity of this work needs large improvement, as the reviewer finds the major contributions and introduction of L2O really hard to follow. Because of the lack of clarity, the reviewer cannot determine the novelty of this work. The proof seems to be reasonable (although the reviewer cannot follow the proof exactly due to the clarity issue), but it seems to be a standard convergence analysis of convex optimizations with different assumptions (Lipschitz continuity, strongly convexity, and some stochasticity). The numerical experiments seem to be reproducible.
>
> A: Thanks reviewer for pointing this out. We have largely revised the paper as suggested.

---

> > ### Comment · Reviewer_5uH7 · 2022-11-16
> > **Response to the rebuttal**
> >
> > Dear Authors,
> >
> > The reviewer appreciates your effort in improving the readability of the manuscript and the reviewer now understands the notations and contributions better.
> >
> > However, based on the contributions themself in this paper, the reviewer would like to maintain the current evaluation, for the two following reasons:
> >
> > 1. **[Heavy notations and lack of theoretical insights]** The reviewer agrees with reviewer 2sEx and Sfir, that this manuscript requires many notations and assumptions (such as strong convexity, the existence of non-trivial optimal solution, Lipschitz properties, and bounded union bounds) to “convexify” the learning objective for achieving an error bound via standard convex optimization techniques. Hence, the reviewer cannot gain new theoretical insights from reading the proof.
> >
> > 2. **[Weak theoretical results and implication]** Even with the heavy assumptions to convexify the optimization objective, the final error bound has an exponential dependence on $Q$ (more precisely, the last term $M_{gT}\alpha\mathcal{O}(TQ^{T-1}{\widetilde{\Delta_{12}}}+Q^{2T-1}\Delta_{12}) $) in theorem 1. If the reviewer understands the setup correctly, $T$ is the number of iterations of the optimizee in each epoch. However, $Q=1+M_{m1}L$ is something larger than 1, which implies that a larger number of iterations of the optimizee in each epoch will result in a large error bound. The reviewer suggests the authors add more clarification to explain the exponential dependency of $Q$ in the final error bound: 1) I.e., in numerical experiments and other larger scales experiments, what is the exact number of $T$? 2)  If such an exponential dependency is inevitable, perhaps explain what is the key component in the current analysis that leads to the exponential dependency, and is there any more practically related assumption to deal with it? With an exponential dependency on $Q$ in the final bound, it is really hard for the practice community to gain insights by reading the proof, which another reviewer (2sEx) has also pointed out the limited implication of this work.
> >
> > On the other hand, as pointed out by other reviewers (egHk and GEXL), it seems that the proposed method is quite novel. Hence the reviewer would also suggest the authors not give up this idea and try to improve the current manuscript, which would potentially lead to a larger impact on the community.

---

> > > ### Author Response · Authors · 2022-11-18
> > > **Response Part II**
> > >
> > > (Continue)
> > >
> > > - This paper investigates the influence of adaptation tasks on generalization error which has not been studied in meta learning theoretically before to our knowledge. Our theorem explicitly exhibits that training-like adaptation tasks contribute to better generalization error. Such a result is against common intuition and has not been captured theoretically previously.
> > >
> > > [r1] Gradient Descent Provably Optimizes Over-parameterized Neural Networks
> > >
> > > [r2] A Convergence Theory for Deep Learning via Over-Parameterization
> > >
> > > [r3] Generalization of model-agnostic meta-learning algorithms: Recurring and unseen tasks
> > >
> > > [r4] Theoretical Convergence of Multi-Step Model-Agnostic Meta-Learning
> > >
> > > [r5] On the convergence theory of gradient-based model-agnostic meta-learning algorithms
> > >
> > > [r6] Safeguarded learned convex optimization
> > >
> > > [r7] Understanding Deep Architectures with Reasoning Layer
> > >
> > > Q: Even with the heavy assumptions to convexify the optimization objective, the final error bound has an exponential dependence on $Q$ (more precisely, the last term $M_{gT}\alpha\mathcal{O}(TQ^{T-1}{\widetilde{\Delta_{12}}}+Q^{2T-1}\Delta_{12}) $) in theorem 1. If the reviewer understands the setup correctly, $T$ is the number of iterations of the optimizee in each epoch. However, $Q=1+M_{m1}L$ is something larger than 1, which implies that a larger number of iterations of the optimizee in each epoch will result in a large error bound. The reviewer suggests the authors add more clarification to explain the exponential dependency of $Q$ in the final error bound: 1) I.e., in numerical experiments and other larger scales experiments, what is the exact number of $T$? 2) If such an exponential dependency is inevitable, perhaps explain what is the key component in the current analysis that leads to the exponential dependency, and is there any more practically related assumption to deal with it? With an exponential dependency on $Q$ in the final bound, it is really hard for the practice community to gain insights by reading the proof, which another reviewer (2sEx) has also pointed out the limited implication of this work.
> > >
> > > A:  We clarify the proposed questions as follows:
> > >
> > > (1) For the iteration number $T$, we choose $T=20$ for our Lasso, Quadratic, and Rosenbrock tasks.  Actually, in large scale experiments [r1,r2] where neural networks are selected as tasks, a common choice is 5 for $T$. Note that $T=5$ is enough for optimizer convergence and our choice $T=20$ is for better experimental performance.
> > >
> > > (2) In terms of exponential dependency on $Q$, it is inevitable but its influence can be canceled by setting up a small step size $\alpha$. The key components that lead to this dependency are the Lipschitz properties to characterize the L2O loss landscape, e.g. the  Lipschitz term $L_{gT}=\mathcal{O}(TQ^{T-2}+Q^{2T-2})$ for $\nabla_\phi g_T(\phi)$. The reason is due to our nested updated procedure $\theta_{t+1}(\phi)=\theta_t(\phi)+m(\nabla_\theta l(\theta_t(\phi)), \phi) (t=0,1,\ldots,T-1)$ and L2O loss definition $g_T(\phi)=l(\theta_T(\phi))$. If we wanna take the gradient of last update term $g_T(\phi)$ over $\phi$, it requires to compute gradients iteratively for all $t=0,1,\ldots,T-1$ and thus contains the exponential term. A similar example is Multi-step MAML which involves a nested structure for inner updates and its convergence theorem [r3] involves exponential term $(1+\alpha L)^N$ where $N$ denotes the inner iteration number.
> > >
> > > Besides, our Lemma captures the Lipschitz upper bound but in practice, the Lipschitz property for L2O loss landscape might not increase exponentially with iteration number $T$. Hence, in terms of practical assumptions, we can make one to characterize the relationship between L2O loss landscape and iteration number $T$ differently, e.g, $L_{gT}=\mathcal{O}(TQ)$, then it will help to relieve the exponential term.
> > >
> > > We also have to emphasize that instead of introducing new assumptions, the influence of exponential dependence on $Q$ can be totally canceled when we set a small determined step size $\alpha$ as stated in Remark 1. Note that the introduction of small step size is quite common in optimization works [r3, r4] when we expect a small error bound.
> > >
> > > [r1] Symbolic Learning to Optimize: Towards Interpretability and Scalability
> > >
> > > [r2] Learning a Minimax Optimizer: A Pilot Study
> > >
> > > [r3] Theoretical Convergence of Multi-Step Model-Agnostic Meta-Learning
> > >
> > > [r4] Generalization of model-agnostic meta-learning algorithms: Recurring and unseen tasks

---

> > > > ### Comment · Reviewer_5uH7 · 2022-11-19
> > > > **Thanks for the reponse**
> > > >
> > > > Dear authors,
> > > >
> > > > The reviewer truly appreciates your effort in defending this work and clarifying the contributions. The reviewer is well aware of the different types of assumptions (Lipschitzness, strong convexity, etc.,) that appear in this submission and prior papers. The reviewer also agrees they are standard assumptions, which allow one to apply certain types of convergence analysis techniques developed using convex optimization (with some stochasticity). The reviewer apologizes if the word "convexify" confuses the authors, it means the major techniques for proving a convergence require some standard assumptions (Lipschitzness, strong convexity, etc.,) that many ML theory papers adopt.
> > > >
> > > > The main reason for the reviewer to reject this paper is **not about the contributions**, is about the **presentation**. For example, [r1] is a good example for people (like the reviewer) who are less familiar with the topic of L2O but know a little bit of ML theory to understand the contributions and implications better.
> > > >
> > > > Some specific suggestions for writing:
> > > > 1. When reading Assumptions 1 & 2 from [r1], the reviewer will directly know these assumptions are standard without asking further questions, perhaps in the next update, the author can try to improve the statement of assumptions in a more reader-friendly way.
> > > > 2. It would be much better for other people to understand the main idea of the L2O problem setting itself if the author can provide a more intuitive introduction to the problem, just like figure 1 in [r1].
> > > > 3. The problem formulation section can be further improved. The reviewer does not have clear suggestions on how to improve the problem formulation section, but Section 2 of [r1] provides a good example of a theoretical framework for studying the meta-learning problem.
> > > > 4. It would also be nice to incorporate "why the exponential dependence is inevitable" and "how $T$ varies in different experiments" and provide a **non-technical remark** on how the theoretical results connect to practice, so that the reviewer (and potentially other readers) can understand what are the practical implications of this work.
> > > >
> > > > References:
> > > > [r1] Generalization of Model-Agnostic Meta-Learning Algorithms: Recurring and Unseen Tasks
> > > >
> > > > In summary, the reviewer certainly believes this submission contains non-trivial theoretical contributions, and the reviewer would be happy to accept the manuscript **if it was submitted in a more reader-friendly way**. Reviewer Sfir also raised a similar issue in the readability, perhaps reviewer 5uH7 is just particularly hard to deal with.

---

> > > > > ### Author Response · Authors · 2022-11-19
> > > > > **Further Response**
> > > > >
> > > > > We thank you for appreciating "this submission contains non-trivial theoretical contributions", and we address the writing suggestions as follows:
> > > > >
> > > > > Q: When reading Assumptions 1 & 2 from [r1], the reviewer will directly know these assumptions are standard without asking further questions, perhaps in the next update, the author can try to improve the statement of assumptions in a more reader-friendly way.
> > > > >
> > > > > A: We believe our assumptions are quite reader-friendly and we demonstrate here: (i) Assumption 1 makes a simple and straight statement about strong convexity, which is quite clear to understand. (ii) Assumption 2 claims the existence of non-trivial meta optimal points. To clarify this assumption, we have carefully rewritten the definition of non-trivial and related equations. Furthermore, we even introduce a clear MAML setting to explain the practical meaning of Assumption 2. The clear clarification has been well recognised by Reviewer GEXL and it is definitely reader-friendly. (iii) Assumption 3 and assumption 4 are about Lipschitz and union bound. Note that the writing of these assumptions are almost exactly the same as Assumption from [r1] which the reviewer believes is in a reader-friendly way.
> > > > >
> > > > > Q: It would be much better for other people to understand the main idea of the L2O problem setting itself if the author can provide a more intuitive introduction to the problem, just like figure 1 in [r1].
> > > > >
> > > > > A: We have added a new figure in Section 3.1 to illustrate the L2O problem setting as suggested in the revision. Such a figure clearly explains the pipeline of L2O problems in an intuitive way.
> > > > >
> > > > > Q: The problem formulation section can be further improved. The reviewer does not have clear suggestions on how to improve the problem formulation section, but Section 2 of [r1] provides a good example of a theoretical framework for studying the meta-learning problem.
> > > > >
> > > > > A: We have reword largely in the problem formulation section in the revision. Specifically, we involve the figure to illustrate the L2O problem which improves the readability a lot. If the reviewer can make more specific suggestions, we are more than happy to revise accordingly.
> > > > >
> > > > > Q: It would also be nice to incorporate "why the exponential dependence is inevitable" and "how $T$ varies in different experiments" and provide a non-technical remark on how the theoretical results connect to practice, so that the reviewer (and potentially other readers) can understand what are the practical implications of this work.
> > > > >
> > > > > A: In terms of incorporation, we have included "why the exponential dependence is inevitable" after Remark 1 and included “how $T$ varies in different experiments" in the Section 5.1 in the revision. In terms of non-technical remark to connect theoretical results with practice, we have involved such discussion in the last paragraph in Section 5.3 in the revision. Specifically, we explain how training-like adaptation implies better L2O generalization in a non-technical way.
> > > > >
> > > > > Finally, we thank the reviewer again for the helpful comments and suggestions for our presentation. If our response resolves your concerns to a satisfactory level, we kindly ask the reviewer to consider raising the rating of our work. Certainly, we are more than happy to address any further comments that you may have.
> > > > >
> > > > > [r1] Generalization of Model-Agnostic Meta-Learning Algorithms: Recurring and Unseen Tasks

---

> > > > > > ### Comment · Reviewer_5uH7 · 2022-11-19
> > > > > > **Thanks and score has been raise.**
> > > > > >
> > > > > > The reviewer is impressed by the dedication and improvement in this submission. Thanks for your effort in the improvement to accommodate a reviewer who is not familiar with L2O optimization. The score has been raised. Some minor suggestions on the typos at the first glance:
> > > > > >
> > > > > > 1. "No that" after equation (6) should be "Note that"
> > > > > > 2. "wanna" in the paragraph after remark 1 is not a formal word, perhaps it should be updated
> > > > > >
> > > > > > There might also be other minor typos as well, but these typos are totally understandable as the reviewer has asked too much in a short period. In the final submission (assuming the paper will eventually get accepted), perhaps the authors can proofread and correct the remaining typos (if there are any). Good luck.

---

> > > > > > > ### Author Response · Authors · 2022-11-19
> > > > > > > **Many thanks for your further updates!**
> > > > > > >
> > > > > > > We thank the reviewer very much for further reviewing our response and raising the score! We will carefully proofread and correct the typos in the final version if accecpted.

---

> > > ### Author Response · Authors · 2022-11-18
> > > **Response Part I**
> > >
> > > We thank you for appreciating “the proposed method is quite novel”, but we have to politely disagree on your assessment of the theoretical strength and insight offered by our work, which we still find potential misunderstandings in your latest comments. Please see below:
> > >
> > > Q: [Heavy notations and lack of theoretical insights] The reviewer agrees with reviewer 2sEx and Sfir, that this manuscript requires many notations and assumptions (such as strong convexity, the existence of non-trivial optimal solution, Lipschitz properties, and bounded union bounds) to “convexify” the learning objective for achieving an error bound via standard convex optimization techniques. Hence, the reviewer cannot gain new theoretical insights from reading the proof.
> > >
> > > A: Let us start by noting that for “learning to optimize”, the theory studies need to look at both “learning”/generalization (from meta training) and “optimization” (from meta testing). This has to be more sophisticated from canonical optimization analysis alone, complicating the problem while standing out our merits. Next, we clarify each assumption here and hope you to re-examine: none is special, nor overly strong.
> > >
> > > - Strong convexity: A majority of theoretical papers [r1, r2] have demonstrated that the loss landscape in the over-parameterized regime exhibits the strong-convexity-like property such as gradient dominance conditions, and the recent strong MAML generalization analysis [r3] also requires such an assumption. Furthermore, we only adopt this assumption in the meta-learning training task’s (or, the “learning” side proof of L2O) generalization analysis. Note that when we study the transferability of the meta-learned point, the “optimization” side proof (applying a trained L2O to a new optimization problem) is free of the strong convexity requirement and applicable for non-convex scenarios.
> > >
> > >
> > > - The existence of the non-trivial optimal solution: As we stated before, we include this assumption to distinguish between MAML and transfer learning. This assumption has been validated in experiments where we expect meta learning helps to learn a well-adapted point rather than optimal point in a single training task. Without such an assumption, our learning objective will be a transfer learning objective rather than a MAML one. Different from what you understood as, $\textbf{this assumption is not related with “convexify” at all}$. As agreed by reviewer GEXL, the assumption is “sensible and standard (i.e., there exists non-trivial optimal points)”
> > >
> > >
> > > - Lipschitz properties and bounded union bounds: These assumptions are rather common and often necessary in the theoretical optimization works [r2, r3, r4, r5, r6] for generic algorithm analysis. One of our theoretical contributions is to firstly characterize generalization for $\textbf {a generic class}$ of L2O algorithms where these assumptions are necessary to capture the general L2O loss landscape. Furthermore, Lipschitz and bounded union bound capture the shape and difference of landscape but do not convexify learning objectives theoretically.
> > >
> > >
> > > As we already discussed: (1) all mentioned assumptions are very standard to characterize meta-training generalization theoretically; (2) when it comes to “optimization” side (testing), our L2O theoretical results are developed without the strongly convex assumption at all and can be claimed in non-convex scenarios. Hence, we politely point out that your latest comment still seems to mis-position our theory  and assumption, e.g., “convexify” the learning objectives and else.
> > >
> > > We also disagree on the point that “reviewers cannot gain new theoretical insights from reading the proof” - hopefully, you can already see new points after we clarify the above. We summarize the important new theoretical insights of this work again for you.  As theory/optimization researchers ourselves, we politely yet confidently defend them:
> > >
> > > - We firstly characterize generalization for a generic class of L2O theoretically. The previous L2O theoretical works [r6, r7] only capture the convergence of L2O or generalization of L2O in restricted settings. Instead, our theorem is applicable for general L2O algorithms transferability study. We further compare the generalization performance between L2O Transfer Learning and M-L2O based on the proposed theorem in Remark 2 which demonstrates superiority of M-L2O.
> > >
> > > - We incorporate MAML results into our meta-learning analysis. The previous MAML theory is developed on the original loss landscape. We further introduce and adapt it into our meta learning part analysis.

---

> ### Author Response · Authors · 2022-11-15
> **Response Part I**
>
> Many thanks for your review!
>
> Q1: The L2O part problem definition needs more descriptions – in section 3.1, it would be much better to introduce the different notations with better characterizations: (a) It would be better to provide more description on the relationship between $\phi$ and $\theta_t$. (b) Is $\xi_j$ a vector data sample or it is a parameter that indexed the task number? (c) What is $z_t$? (d) In the sentence above equation, $\zeta_t$ is referred to as a data sample, while in the first sentence of section 3.1, $\xi_t$ is referred to as a data sample, what is the relationship between  $\zeta_t$ and $\xi_t$? (e) Where does equation 2 come from? Perhaps the author can provide more introduction/motivation on why $\nabla_\phi \theta_T(\phi)$ has the form in equation (2)?
>
> A: Thanks for the comments. We have rewritten the description in Section 3.1 in the revision.
>
> (a) $\phi$ denotes the optimizer parameter, $\theta_t$ denotes the optimizee parameter in $t$-th iteration. In L2O, instead of adopting SGD or ADAM to update $t$-th iteration neural network parameter $\theta_t$, we introduce LSTM or MLP model which takes the parameter gradient as input $z_t$ for update computation. We call such an LSTM or MLP model $\textbf{optimizer}$ since it plays the same role as the traditional optimizer. Note that we call the regular loss function $\textbf{optimizee}$. Specifically, the optimizer model is parameterized by $\phi$. In conclusion, $\theta_t$ is updated by the optimizer hence it is the function of the optimizer parameter $\phi$.
>
> (b) $\xi_j$ is a vector data sample and it denotes the $j$-th sample of a data batch of shape $N$.
>
> (c) $z_t$ is the input of the optimizer model. Typically, we take the current iteration optimizee parameter gradient as $z_t$, i.e., $z_t=\nabla_\theta \hat{l}(\theta_t(\phi))$.
>
> (d) We only adopt $\xi_j$ notation. To clarify the difference between $\xi_j$ and $\zeta_t$, $\xi_j$ refers to the $j$-th single sample in the whole data batch of size $N$. Instead, $\zeta_t$ refers to a data batch of size $N$ used in $t$-th iteration while there are totally $T$ iterations. We make this comment in the revision.
>
> (e) Equation 2 calculates the optimizee parameter $\theta_T(\phi)$ gradient over the optimizer parameter $\phi$ in the $T$-th iteration. The detailed proof is in the Lemma1 in Appendix A.1. Based on parameter update procedure $\theta_{t+1}(\phi)=\theta_t(\phi)+m(\nabla_\theta \hat{l}(\theta_t(\phi)), \phi)$, it is observed that each $t$-th iteration update $m(\nabla_\theta \hat{l}(\theta_t(\phi)), \phi)$ is the function of $\phi$. Hence, if we compute $T$-th parameter $\theta_T(\phi)$ gradient over $\phi$, it will be the summation of last $T$ iterations gradients. Meanwhile, each iteration gradient takes the production form as shown in Equation 2.
>
> Q2:The intuition of M-L2O is not clearly stated – why is the empirical risk (equation 5) related to MAML?
>
> A: The meta objective in MAML[1] is defined as $\sum_{i} f_i(\theta-\alpha\nabla_\theta f_i(\theta))$ which takes the average of each individual task gradient to update the outer task. Such a method enables the network to locate a well-adapted initial point. Hence, we propose Equation 5 that $\hat{G}_T(\phi)=\hat{g}_T(\phi-\alpha\nabla_\phi\hat{g}_T(\phi))$ as meta objective for L2O optimizer. We expect to find a well-adapted initial optimizer point by this meta empirical risk.
>
> Q3: The notations in section 3.2 are confusing – where do $g_T^1, g_T^2, g_T^3$ appear in Algorithm 1?
>
> A: Thanks for pointing this out. We have revised Algorithm 1. In conclusion, we meta-train the optimizer on training task $g_T^1$ and get the optimizer initial point   $\widetilde{\phi}^1_{MK}$. We then further update optimizer with adaptation task $g_T^2$
> and finally apply the updated optimizer to train a new optimizee task $g_T^3$.
>
> [1]: Model-Agnostic Meta-Learning for Fast Adaptation of Deep Networks

---

### Official Review · Reviewer_GEXL · 2022-10-24

**Confidence:** 3
**Correctness:** 3
**Technical Novelty And Significance:** 3
**Empirical Novelty And Significance:** 2
**Recommendation:** 8

**Clarity, Quality, Novelty And Reproducibility:**

Writing quality is mediocre, with a number of obvious typos such as “a out-of-distibution”, etc.

Novelty is solid but also limited in some way, since the idea appears to combine two existing ideas: L2O and MAML.

No code was attached. But experiments seem straightforward and should be reproducible.


**Strength And Weaknesses:**

Strength:
-	This paper tackles a very important problem and inherent limitation of L2O: its performance is not ensured on tasks different from those seen in training. If resolved, that can remove a notable hurdle for L2O in practice.
-	The authors formulated an MAML-like nested optimization, to locate in well-adapted region where a few adaption steps enable optimizer to generalize well on unseen tasks. The formulation is solved by meta-training with theory backups.
-	Theoretical analysis reveals that training-like adaptation can mean better generalization, which experiment results support. This is a meaningful new insight.

Weakness:
-	It is hard for me to catch whether/what the main innovations are for the theory part. The idea seems to be a direct combination of MAML and L2O, both belonging to the meta learning family.
The authors are invited to elaborate more on: is their theoretical result reused/re-instantiated from some known meta-learning result? Or by some direct combination of MAML result and L2O result each? Or they have actually made noteworthy theory contributions?
I believe the clarity of Section4 will benefit a lot by adding such discussions. Currently, I find it very difficult to assess the authors’ theoretical contributions as those are poorly contextualized. The whole Section 4 surprisingly has not cited or discussed a single theory paper!

-	There are existing L2O works studying generalization by theory, such as (Chen 2020d). However, the authors fail to concretely discuss how their generalization results differ from/compare with Chen et. al.
Another relevant paper the authors fail to cite and discuss is: “safeguarded learned convex optimization”, Heaton et. al.

-	The experiments are in general consistent and informative, but unfortunately on very simple test problems only. I would expect at least some constrained optimization, nonconvex optimization or NNs. The authors are encouraged to add some such result. This could be considered as a minor point, IF the authors are able to clarify their theoretical contribution to be major (rather than combining existing off-the-shelf ingredients).


**Summary Of The Paper:**

Learning to Optimize (L2O) has emerged as a data-driven way to derive “learned optimizers” for specific problem classes. However, for very different unseen problems those learned optimizers can fail badly. This paper is the first to discuss L2O’s test-time self-adaptation to out-of-distribution tasks, providing both theoretical and empirical results.

**Summary Of The Review:**

In summary, I think this paper targets a very important problem and critical limitation of L2O, that will improve the applicability of L2O. The proposed solution seems reasonable with a well defined nested optimization problem, and the associated theoretical analysis well justify the motivation of the proposed solution.

There are some minor concerns in the "Weakness" part, if the authors can well address my concerns I would be able to change my score.

---

> ### Author Response · Authors · 2022-11-15
> **Response Part II**
>
> Many thanks for your expert review!
>
> Q2: The experiments are in general consistent and informative, but unfortunately on very simple test problems only. I would expect at least some constrained optimization, nonconvex optimization or NNs. The authors are encouraged to add some such result. This could be considered as a minor point, IF the authors are able to clarify their theoretical contribution to be major (rather than combining existing off-the-shelf ingredients).
>
> A: As suggested, we have additional experiments on a non-convex optimization problem with Rosenbrock [1] loss functions as the objectives of the optimizees. The Rosenbrock optimizees are used for both adaptation and testing. We repeat the experiments for 10 times, and we collect the mean of the logarithm of the objective values and their standard deviation at 500-th step in the following table.
>
>  | Method | Mean Log Objective | Standard Deviation |
> | :------: | :------: | :------: |
> | Vanilla L2O | 0.977 | 0.225 |
> | DT | -4.864 | 0.395 |
> | TL | -2.170 | 1.312 |
> | M-L2O | -6.832 | 0.445 |
>
> We can observe that M-L2O clearly outperforms other baselines methods by significant margins (smaller log loss value by over 1.9), which highlights the effectiveness of our proposals. We have also added these experiment analysis to Appendix A.5 in the revision.
>
>
> Q3: Writing quality is mediocre, with a number of obvious typos such as “a out-of-distibution”, etc.
>
> A: Thanks for your comment. We have revised the paper and corrected typos as suggested.
>
> [1] An automatic method for finding the greatest or least value of a function

---

> > ### Comment · Reviewer_GEXL · 2022-11-16
> > **Appreciate the authors' dedicated response, and I will raise my score to 8.**
> >
> > I would like to thank the authors for the thoughtful answers. I have also read through other reviews / responses.
> >
> > Initially, the authors seem to have made overly strong assumptions, and the theoritical originality was also unclear to me. The authors responded by clarifying (1) the assumption 2 is sensiable and standard (i.e., there exists non-trivial optimal points); (2) their generalization result is much broader than existing L2O works. The authors should include those clarifications into their paper.
> >
> > The paper has improved clarity after revision too. I also like the fact that it discussed the important unseen optimizee generalization issue for the first time, regarding general L2O methods (in contrast to a problem-specific formulation).
> >
> > Overall, the method is solid, and the novelty is now clarified. I think it makes good contributions to L2O theory and practice. Hereby I'm raising my score to 8.

---

> ### Author Response · Authors · 2022-11-15
> **Response Part I**
>
> Many thanks for your expert review!
>
> Q1 : It is hard for me to catch whether/what the main innovations are for the theory part. The idea seems to be a direct combination of MAML and L2O, both belonging to the meta learning family. The authors are invited to elaborate more on: is their theoretical result reused/re-instantiated from some known meta-learning result? Or by some direct combination of MAML result and L2O result each? Or they have actually made noteworthy theory contributions? I believe the clarity of Section4 will benefit a lot by adding such discussions. Currently, I find it very difficult to assess the authors’ theoretical contributions as those are poorly contextualized. The whole Section 4 surprisingly has not cited or discussed a single theory paper!
>
> ```
> There are existing L2O works studying generalization by theory, such as (Chen 2020d). However, the authors fail to concretely discuss how their generalization results differ from/compare with Chen et. al.
> ```
> Another relevant paper the authors fail to cite and discuss is: “safeguarded learned convex optimization”, Heaton et. al.
>
> A: Thanks for such comments. We cited all mentioned works and added up a new discussion in Section 4 to clarify the difference between our theory and existing works. Our theoretical work can be characterized into two parts: L2O area and meta learning area.
>
> In terms of L2O area, the previous work [3] only analyzed the worst-case convergence of proposed safe-L2O but not generalization; while another work [4] analyzed the generalization of quadratic-based L2O (named “reasoning” layer in their paper). Instead, we develop the generalization results on a general class of L2O problems.
>
> In terms of meta-learning area, the previous works have demonstrated the convergence and generalization of MAML [2, 5]. We firstly apply such MAML results to measure the distance between convergent point and optimal point in the meta training task. Note that this analysis is different from our newly developed L2O analysis where we characterize transferability of the training learned point on meta testing tasks. Then, we develop the theorem based on L2O and meta learning results. Our corollary further demonstrates that training-like adaptation tasks can contribute to better generalization in L2O, which is against common intuition.
>
> In short, our theoretical novelty lies in three aspects:
> - Firstly theoretically characterize generalization for a generic class of L2O algorithms
> - Incorporating the MAML results in our meta learning analysis.
> - We theoretically prove that both training-like and testing-like adaptation contribute to better generalization in L2O.
>
> In the mentioned theory work (Chen 2020d) [4], the author developed the generalization theory under quadratic optimization and its generalization error measures the gap between expected loss and empirical loss. Instead, we develop the generalization theorem without problem formulation restriction and our generalization error is determined by testing task performance.
>
> In the other mentioned work of Heaton [3], the author developed the convergence theorem of a fixed-point algorithm by resorting to first-order method on a convex problem. Instead, we develop a generalization theorem without first-order method and algorithm restriction.
>
> [1] An automatic method for finding the greatest or least value of a function
>
> [2] Generalization of model-agnostic meta-learning algorithms: Recurring and unseen tasks
>
> [3] Safeguarded learned convex optimization
>
> [4] Understanding Deep Architectures with Reasoning Layer
>
> [5] Theoretical Convergence of Multi-Step Model-Agnostic Meta-Learning

---

### Official Review · Reviewer_egHk · 2022-10-25

**Confidence:** 3
**Correctness:** 2
**Technical Novelty And Significance:** 3
**Empirical Novelty And Significance:** 2
**Recommendation:** 6

**Clarity, Quality, Novelty And Reproducibility:**

Low writing clarity (missing lots of details, see above). Reproducibility is also low in the current shape due to the important experimental detail absence. I look forward to the authors’ clarifications.



**Strength And Weaknesses:**

This paper does seem to have merits, but they are largely compromised by the unclarity in the current manuscript. See below:

Strength.
- The unseen generalization is an open L2O challenge that was unexplored before. The authors demonstrate a novel solution that outperformed naïve baselines, such as direct transfer learning.
- The authors presented theoretical evidence that their meta-adaption design grants M-L2O optimizer faster adaption ability for out-of-distribution tasks and can have smaller generalization errors, compared to vanilla L2O.

Weakness & Questions.
- It is unclear to me whether the theoretical assumptions made in Section 4.1 make sense or not for L2O. As far as I know, L2O adopts an LSTM to predict the update. How can an LSTM g function be strongly convex? Also, the Lipschitz condition can easily get trivial under recurrence.
- Experiments are short of MANY details. For example, nowhere in the main text or supplement, did the authors report what L2O algorithm they actually used for all experiments! That shows the paper was finished in a hectic rush, lacking serious proofreading and also making the comparison fairness questionable.
- The authors conclude that a training-like optimizer adaptation task outperforms a test-like one. That is non-intuitive, and I’m not fully convinced: what if we smoothly interpolate the parameters from training to testing, in a curriculum-learning way? I also suspect Figure 3 is because training/testing cases are still quite similar.
- The optimization problems tested are too simple: only Lasso and quadratic.



**Summary Of The Paper:**

The paper investigates the generalization of L2O on unseen test cases that substantially differ from the training distribution. It proposes a self-adapted L2O algorithm (M-L2O) incorporated with meta-adaptation. The generalization advantages of M-L2O over out-of-distribution tasks have been theoretically and empirically validated.

**Summary Of The Review:**

Please refer to the comments on strengths and weaknesses.

---

> ### Author Response · Authors · 2022-11-15
> **Many thanks for your expert review**
>
> Many thanks for your expert review!
>
> Q1: It is unclear to me whether the theoretical assumptions made in Section 4.1 make sense or not for L2O. As far as I know, L2O adopts an LSTM to predict the update. How can an LSTM g function be strongly convex? Also, the Lipschitz condition can easily get trivial under recurrence.
>
> A: Given what the current deep learning theory techniques can handle, the assumptions made in Section 4.1 are standard and yet still capture the local geometry of typical neural network loss functions. For the strongly-convex assumption, many theoretical papers [1,2] have demonstrated that the loss landscape in the over-parameterized regime exhibits the strong-convexity-like property such as gradient dominance conditions. Furthermore, the latest MAML generalization analysis [3] also requires such an assumption. For the Lipschitz conditions, they have been widely assumed in recurrence optimization papers [3, 4]. It is still not trivial since our analysis (Theorem 1) still characterizes how the basic Lipschitz term ($Q=1+M_{m1}L$) affects the generalization error under recurrence.
>
> Q2: Experiments are short of MANY details. For example, nowhere in the main text or supplement, did the authors report what L2O algorithm they actually used for all experiments! That shows the paper was finished in a hectic rush, lacking serious proofreading and also making the comparison fairness questionable.
>
> A: The L2O algorithm we used for all experiments is RNNProp-CL introduced in [6]. The comparison between methods is fair as we apply the same settings for training, adaptation and testing for all experiments. We have included all experimental details in the updated Section 5 and Appendix A.5. Furthermore, we also update the figure with error bars which also demonstrate that our M-L2O enjoys a smaller variance across different seeds.
>
> Q3: The authors conclude that a training-like optimizer adaptation task outperforms a test-like one. That is non-intuitive, and I’m not fully convinced: what if we smoothly interpolate the parameters from training to testing, in a curriculum-learning way? I also suspect Figure 3 is because training/testing cases are still quite similar.
>
> A: Thanks for the questions. Both our experiments and theory demonstrate that both training-like and testing-like adaptation tasks can help to improve generalization performance. Our conclusion that training-task performs better is totally based on current experimental results and we aim to let the community notice such a phenomenon. We acknowledge that testing-like adaptation tasks perform better in lots of existing meta-learning settings.
>
> We further implement the interpolation experiments as suggested and include the results in Appendix A.5. In Figure A4 (b), smaller $\alpha$ refers to more testing-like adaptation tasks while larger $\alpha$ represent training-like ones. Both TL and M-L2O results validate our claim that training-like optimizer adaptation tasks perform better than testing-like tasks.
>
> Q4: The optimization problems tested are too simple: only Lasso and quadratic.
>
> A: We have conducted additional experiments on a canonical non-convex optimization problem Rosenbrock [5] and present the results in Appendix A.5 in the revised paper. We train the optimizers on the LASSO optimizees, and perform adaptation and testing on the Rosenbrock optimizees. Each experiment is repeated 10 times, and the results at 500th step (both the mean and the standard deviation of the log objective) are collected and presented in the table below:
>
> | Method | Mean Log Objective | Standard Deviation |
> | :------: | :------: | :------: |
> | Vanilla L2O | 0.977 | 0.225 |
> | DT | -4.864 | 0.395 |
> | TL | -2.170 | 1.312 |
> | M-L2O | -6.832 | 0.445 |
>
> The new experiments show that M-L2O achieves a superior performance compared with other benchmarks (DT, TL, Vanilla L2O) on the non-trivial non-convex optimization problem.
>
>
> [1] Gradient Descent Provably Optimizes Over-parameterized Neural Networks
>
> [2] A Convergence Theory for Deep Learning via Over-Parameterization
>
> [3] Generalization of model-agnostic meta-learning algorithms: Recurring and unseen tasks
>
> [4] Theoretical Convergence of Multi-Step Model-Agnostic Meta-Learning
>
> [5] An automatic method for finding the greatest or least value of a function
>
> [6] Training Stronger Baselines for Learning to Optimize

---

> > ### Comment · Reviewer_egHk · 2022-11-16
> > **Reply to the rebuttal**
> >
> > Thanks for the detailed response. The clarification helps a lot, and the paper's readability benefits remarkably from the added details as well as the new nonconvex test case.
> >
> > Overall, the paper seems to have made a worthy and novel contribution both theoretically and practically, i.e., improving out-of-distribution test case generalization, to the L2O field. The rebuttal also well addressed most of my previous concerns. Thus, I raise my rating to 6 and tend to support its acceptance.

---

### Comment · Area_Chair_PoCz · 2022-11-20
**Please update your reviews**

Please make sure that your reviews acknowledge authors’ responses and reflect your current evaluation of the paper. This is particularly important if you didn’t directly engage with the authors during the discussion phase (so the authors don’t know if their response changed your evaluation) or if you expressed an intention to update your rating but did not do so.

Cheers,
AC

---

### Decision · Program_Chairs · 2023-01-20

**Decision:**

Accept: poster

**Justification For Why Not Higher Score:**

Writing quality is limited even after revision; experiments are only on small models and datasets hence it’s unclear how practically impactful the work will turn out to be.



**Justification For Why Not Lower Score:**

This paper tackles a very important problem and inherent limitation of L2O: out-of-distribution task generalization. This is hardly touched by prior L2O arts, and if resolved, it can remove a notable hurdle for L2O in practice. The authors’ theoretical analysis looks solid, and the numerical experiments corroborate the theoretical results. Hence despite still preliminary in many ways, the paper is already self-contained and has clear merits to be accepted.



**Metareview: Summary, Strengths And Weaknesses:**

Learning to Optimize (L2O) can efficiently solve problems similar to those seen in training, but often struggles when new test problems come with a substantially deviation from the training task distribution. This paper looks into this open challenge, by meta-training an L2O optimizer that can perform fast test-time self-adaptation to an out-of-distribution task. The authors provides theoretical analysis into their L2O’s provably better generalization than vanilla ones. They demonstrate empirical on some standard optimization problems.

All reviewers initially agreed on the novelty and contribution in this paper but made various critiques on the paper’s clarity, including both theoretical assumptions and experimental details. Indeed, the paper’s initial shape wasn’t great. But the authors then demonstrated truly remarkable dedication and improvement during the discussion period, for which the AC applauds. They submitted their new thoroughly revised draft too. After extensive open forum discussions, four out of five reviewers raised their score, including one substantial increase from 3 to 8. Hence, the AC concludes that the paper quality becomes satisfactory and deserves acceptance. The authors are instructed to proofread their paper times and again, better with the help of a native speaker.


**Note From Pc:**

if the above contains the word "oral" or "spotlight" please see: "oral" presentation means -> notable-top-5% and "spotlight" means -> notable-top-25%. As stated in our emails, we are disassociating presentation type from AC recommendations